


# Sources of organic ice nucleating particles in soils

T. C. J. Hill[1], P. J. DeMott[1], Yutaka Tobo[2,3], J. Fröhlich-Nowoisky[4], B. F. Moffett[5], G. D. Franc[6#] and S. M. Kreidenweis[1]

[1]Department of Atmospheric Science, Colorado State University, Fort Collins, CO, 80523, USA.
[2]National Institute of Polar Research, Tachikawa, Tokyo 190-8518, Japan. Japan.
[3]Department of Polar Science, School of Multidisciplinary Sciences, SOKENDAI (The Graduate University for Advanced Studies), Tachikawa, Tokyo 195-8518, Japan.
[4]Multiphase Chemistry, Max Planck Institute for Chemistry, Mainz, 55020, Germany.
[5]Ocean Lab, Goodwick, SA64 0DE, United Kingdom.
[6]Plant Sciences Department, University of Wyoming, Laramie, WY, 82071, USA.
[#]Deceased

*Correspondence to*: T. C. J. Hill (Thomas.Hill@colostate.edu)

**Abstract.** Soil organic matter (SOM) may be a significant source of atmospheric ice nucleating particles (INPs), especially of those active >-15 °C. However, due to both a lack of investigations and the complexity of the SOM
itself, the principal sources of these INPs remain unknown. To more comprehensively characterize organic INPs we tested locally representative soils in Wyoming and Colorado for total organic INPs, INPs in the heat-labile fraction, ice nucleating (IN) bacteria, IN fungi, IN fulvic and humic acids, IN plant tissue, and ice nucleation by monolayers of aliphatic alcohols. All soils contained ≈$10^6$ to ≈5 ×$10^7$ INPs g$^{-1}$ dry soil active at -10 °C. Removal of SOM with $H_2O_2$ effectively removed all INPs active >-18 °C (the limit of testing), while heating of soil suspensions to 105 °C showed
that labile INPs increasingly predominated >-12 °C and comprised ≥90% of INPs active >-9 °C. Papain protease, which inactivates IN proteins produced by the fungus *Mortierella alpina*, common in the region's soils, lowered INPs active at ≥-11 °C by ≥75% in two arable soil and sagebrush shrubland soil. By contrast, lysozyme, which digests bacterial cell walls, only reduced INPs active at ≥-7.5 or ≥-6 °C, depending on the soil. The known IN bacteria were not detected in any soil, using PCR for the *ina* gene that codes for the active protein. We directly isolated and
photographed two individual INPs from soil, using repeated cycles of freeze-testing and subdivision of droplets of dilute soil suspensions: They were complex and apparently organic entities. Ice nucleation activity was not affected by digestion of Proteinase K-susceptible proteins or the removal of entities composed of fulvic and humic acids, sterols or aliphatic alcohol monolayers. Organic INPs active at temperatures colder than -10° to -12 °C were resistant to all investigations other than heat, oxidation with $H_2O_2$ and, for some, digestion with papain. They may originate
from decomposing plant material, microbial biomass and/or the humin component of the SOM. If the latter then they are most likely to be a carbohydrate. Reflecting the diversity of the SOM itself, soil INPs have a range of sources, occur with differing relative abundances, and may be protected by different mechanisms.





# 1 Introduction

We have known for more than four decades that soils are reservoirs of organic ice nucleating particles (INPs) (Vali, 1968; Schnell and Vali, 1972; 1973; 1976). Decaying leaf litter was found to produce up to $10^{10}$ INPs per gram active at -10 °C,

and with an onset of activity at up to ≈-1.5 °C (Schnell and Vali, 1972; 1973). Rich top soils and peat were shown to be ice nucleating (IN) at -4 °C (Vali, 1968). The component responsible for the highest temperatures of freezing, an IN bacterium, was soon isolated (Maki et al., 1974; Vali et al., 1976). However, the proposal (Schnell and Vali, 1972) that particles composed of soil organic matter (SOM) may act as INPs was contested (Rosinski et al., 1974) and then neglected as a topic for investigation, even though the aerial release of INPs from litters (Schnell and Vali, 1976) suggested that this may be a

significant atmospheric source, especially of INPs active ≥-10 °C. INPs influence cloud microphysical and radiative properties, the release of latent heat, and the triggering of precipitation.

Recently there has been a resurgence of related research, with the general aims of correlating abundances of organic INPs in the boundary layer with local ecotypes (Prenni et al., 2009; Bowers et al., 2010; Garcia et al., 2012; Huffman et al., 2013;

Tobo et al., 2013; Mason et al., 2015), gauging the stimulation of their release by humidity, rainfall and harvesting (Garcia et al., 2012; Huffman et al., 2013; Prenni et al., 2013; Bigg et al., 2015; Wright et al., 2015), and measuring organic INP abundance in clouds and precipitation (Christner et al., 2008; Delort et al., 2010; Joly et al., 2014; Monteil et al., 2014; Morris et al., 2008; Šantl-Temkiv et al., 2015).

By contrast, limited progress has been made characterizing the sources of these "bio-INPs". Heating to 90-100 °C to deactivate IN macromolecules and digestion with $H_2O_2$ to entirely remove the SOM confirmed their dominance of the INP population active ≈>-15 °C in diverse sieved soils (Conen et al., 2011; O'Sullivan et al., 2011). Further, their removal with heat (300 °C) and $H_2O_2$ digestion revealed that organic-rich predominated over mineral INPs active >-36 °C in soil dusts generated from Wyoming agricultural soils (Tobo et al., 2014). Recent investigations into specific sources of organic INPs

found that the soil-inhabiting fungus *Mortierella alpina*, that is common in many of the soils studied here, is an ice nucleator: all strains tested initiated freezing at -5° to -6 °C and released large numbers of cell-free INPs (Fröhlich-Nowoisky et al., 2015). Of potential widespread relevance, too, is the recent discovery of ice nucleation by cellulose (Hiranuma et al., 2015); activity was detected starting at -16 °C.

To more comprehensively assess the principal sources of organic INPs in soils, we tested soils representative of local ecotypes in Wyoming and Colorado for the presence of INPs from a wide range of potential sources. We tested for total





organic INPs, INPs in the heat-labile fraction, IN bacteria (using PCR of the *ina* gene that codes for the Ina protein (Warren, 1995)), IN *M. alpina* (by digestion with papain protease), IN fulvic and humic acids (via acid and alkaline extractions), ice nucleating entities comprised of crystals of sterols or monolayers of aliphatic alcohols (by removal with chloroform), and IN plant tissue. We also demonstrated that cycles of freeze-testing and subdivision of droplets of dilute soil suspensions can be used to directly isolate individual INPs from soil for subsequent analyses.

## 2 Materials and methods

### 2.1 Soil sampling

Topsoils were sampled from beneath representative natural ecotypes and land uses, mostly in Wyoming. These included conventionally and organically cropped land and pasture at the University of Wyoming's Sustainable Agricultural Research and Extension Center (SAREC), near Lingle, Wyoming, sagebrush shrubland, native glacial scree grassland and lodgepole pine forest in south-east Wyoming, and ponderosa pine forest at the Manitou Experimental Forest in Colorado. Details of sites and vegetation are given in Table 1. At each site, three replicate soil samples were obtained. Each was obtained from a separate $10 \times 10$ m area, and within each area three cores, each 5 cm deep and ≈10 cm in diameter, were retrieved and mixed together on site. Samples were thoroughly mixed and roots removed, and then stored at 4 °C before testing. Sub-samples were frozen at -20 °C for DNA extraction and later analyses.

Dry weights were obtained by heating a sub-sample of each soil to 105 °C for 24 h. For analysis of soil organic carbon and total nitrogen, soil samples with a pH >7 were initially immersed in 0.1 M HCl overnight to remove carbonates and then filtered (No. 42 paper, Whatman) and dried as above. Samples were ground and sieved through a 0.5 mm sieve, and analysed using a CHN analyser.

### 2.2 Measurement of INPs

Concentrations of INPs in soil suspensions were derived using the immersion freezing method. Samples were made initially into a 10-fold slurry with the appropriate INP-free diluent, mixed on an orbital shaker at 200 rpm for 20 min and then a series of 10-fold dilutions constructed. INP estimates were made using 32 aliquots of 50-80 µL of suspension dispensed into sterile 96-well polypropylene PCR trays (Life Science Products Inc.). Details of the immersion freezing protocol and device are given in Garcia et al. (2012). Results were adjusted for INPs in the diluent (INP-free until <-15 °C). The effect of papain digestion was assessed using CSU's ice spectrometer, a method that eliminates the transfer procedures used in the above method (details of the method are given in Hiranuma et al., 2015). Numbers of INPs in soil suspensions were estimated using Eq. (1):



$$INPs\ mL^{-1} = \frac{-\ln(f)}{V}, \tag{1}$$

where $f$ is the proportion of droplets not frozen and $V$ is the volume of each aliquot (Vali 1971), and then converted to INPs $g^{-1}$ of dry soil. Confidence intervals (95%) were derived using the formula (no. 2) recommended by Agresti and Coull
(1998).

## 2.3 Removal of SOM

Removal of SOM was achieved by digestion with hydrogen peroxide. Fifty millilitres of 15% $H_2O_2$ was added to 2.5-10 g fresh soil, the slurry steeped until initial foaming subsided, and then boiled while topping up with $H_2O_2$ as required until all SOM and then all residual $H_2O_2$ was decomposed. Suspensions were left overnight to settle, the pH restored if necessary and
the volume made up to a 10- dilution with deionized water. Ten-fold serial dilutions were then made with 0.45 μm pore diameter filtered 10 mM sodium phosphate buffer (pH 7) or deionized water (comparison tests showed no difference).

For the sugar beet and organic grass/alfalfa fallow soils, and kaolinite, a comparison of INP concentrations were made for the untreated and $H_2O_2$ digested samples with comparably treated aerosolized soil dusts (0.6 μm dia.) measured using a
continuous flow diffusion chamber (Tobo et al., 2014). INP fraction data presented by Tobo et al. (2104) were converted to per gram values assuming the particles were spherical and had a density of 2.6 g cm$^{-3}$.

## 2.4 Removal of humics

Testing for INPs of humic origin was performed on the SAREC pasture soil since it contained ≈5% SOM and a commensurately high level of INPs. Fulvic and humic acids were removed using the International Humic Substance Society
method as described in Swift (1996). Briefly, 1 M HCl and deionized water were added to a sample until a 10:1 liquid:dry-soil weight ratio until a pH of ≈1.5 was obtained. The suspension was shaken for 1 h, settled, and the lightly coloured supernatant decanted off. The sediment was neutralized with 1 M NaOH and then 0.1 M NaOH was added until a 10:1 liquid:dry-soil weight ratio was reached. This was shaken intermittently for 4 h, settled overnight, and the now dark brown supernatant containing the humic acids removed by decanting. Suspensions of the residual soil were then made in 0.45 μm
pore diameter filtered 10 mM sodium phosphate buffer at a pH value approximating the soil sample and tested for INP content.

## 2.5 Removal of sterols and aliphatics

Two contrasting soils, the organic horizon from the lodgepole pine forest and the sagebrush shrubland soil, were tested for ice nucleating entities sourced from monolayers of alcohols, sterol crystals and other chloroform-soluble organics. This was
assessed by dissolving the organics and testing for any decrease in ice nucleation activity. Soils were air-dried and then ≈5 g





added to 30 mL chloroform, the mix shaken and then steeped for 3 days before removing the chloroform by vacuum filtration using a glass filter holder fitted with a 0.45 µm pore diameter cellulose filter (Gelman). Samples were then rinsed three times with fresh chloroform and dried overnight. Suspensions of the soils were made in buffer (10 mM sodium phosphate buffer at pH values approximating the original soil sample, and filtered through a 0.1 µm pore diameter syringe
filter (Nalgene, Thermo Scientific) and tested for INP content.

### 2.6 Heat treatments

Ice nucleating particles produced by IN bacteria and fungi (probably including lichen symbionts) are typically proteins (see Pummer et al., 2015). Protein function is controlled by secondary and tertiary structure, and in IN bacteria by the formation of complexes (eg, Kozloff et al., 1991; Schmid et al., 1997; Garnham et al., 2011). Heat is a simple but effective tool to
disrupt and denature these proteinaceous INP. Two temperatures, 60° and 105 °C, were used.

Sixty degrees Celsius is a moderate heat that will fragment molecular complexes and denature heat-labile proteins. Accordingly, in tests on diverse IN bacteria isolated from crops and grasses, incubation at 60 °C for 20 min degraded INPs from class A aggregates nucleating ≥-3 °C to class B oligomers typically freezing at -8° to -9 °C (Turner et al., 1990; Hill et
al., 2014), while 60 °C for 60 min deactivated the INPs of the IN fungi *Acremonium implicatum* and *Isaria farinosa* (Pummer et al., 2015) isolated from boundary layer air in Colorado (Huffman et al., 2013). INPs produced by bacteria and fungi are, however, often unaffected by this temperature. This includes leaf-derived ice nuclei (Schnell and Vali, 1973), most strains of IN *M. alpina* (Fröhlich-Nowoisky et al., 2015), IN lichen (Kieft and Ruscetti, 1990), and IN *Fusarium* spp. (Pouleur et al., 92), although the onset temperature of activity of cell-free INPs of *F. avenaceum* was lowered when heated
>40 °C (Hasegawa et al., 1994). To test the sensitivity of soil INPs to moderate heat, PCR trays were thawed after initial INP testing and then heated in the thermal cycler to 60 °C for 20 min and retested.

To test the effect of a more denaturing heat, 1.8 mL of a 100-fold dilution of soil suspension was aliquoted into a 2 mL screw-cap microcentrifuge tube and immersed in mineral oil at 105 °C for 20 min (plus 7 min for equilibration). Serial
dilutions were made with 0.45 µm pore size filtered 10 mM sodium phosphate buffer (at pH values approximating the original soil sample).

### 2.7 Enzymatic digestions

The enzyme lysozyme lyses all bacteria by digesting the cell wall polymer peptidoglycan. It also hydrolyses fungal chitin oligosaccharides, but not the chitin polymer itself. Gram-negative bacteria such as the IN bacteria are less susceptible to
lysozyme digestion of their cell walls than Gram-positives because their outer membrane acts as a barrier and because their cell walls have less of the degraded molecule, peptidoglycan. Their efficient lysis requires EDTA which, by compromising





the integrity of the outer membrane, assists the passage of lysozyme. Tests on pure cultures of diverse IN bacterial isolates from (grown in nutrient broth with cell densities measured using absorbance at 600 nm), confirmed that addition of EDTA was required (optimal at 5 mM) for efficient lysis with lysozyme at 4 mg mL$^{-1}$ (Pooley and Brown, 1990), a concentration comparable to that used by Christner et al. (2008). A series of soil suspensions ranging in dilution from $10^{-2}$ to $10^{-7}$ were

constructed in 0.45 µm pore diameter filtered 10 mM Tris (pH 8), 5 mM EDTA (pH 8) and 4 mg mL$^{-1}$ lysozyme (L7651, Sigma-Aldrich), and incubated at 16 °C for 24 h. The control comprised soil suspensions in Tris and EDTA incubated under the same conditions.

To test soils for the presence of INPs susceptible to digestion by Proteinase K, such as those possessed by IN *Fusarium*

species (Hasegawa et al., 1994; Humphreys et al., 2001), suspensions of the agricultural soils were digested with 5 U mL$^{-1}$ Proteinase K (Qiagen) using the same buffer, incubation conditions and negative controls as used for lysozyme (also optimal for this enzyme). Soil suspensions (1 in 20 dilution) were also digested with papain protease (AppliChem) at concentrations of 10 or 20 mg mL$^{-1}$ in 10 mM sodium phosphate buffer (pH 6.3) at 60 °C for 20 min. Controls were again processed in the same manner except without the enzyme. The INP-free buffer was used for serial dilutions for INP determinations.

Adsorption of enzymes onto the surfaces of minerals, with consequent deactivation of their IN sites, as observed by Zolles et al. (2105), does not appear to have been a factor in the soils tested; lysozyme and Proteinase K had no effect upon INP counts below their expected impacts at relatively warm temperatures.

### 2.8 PCR of the *ina* gene in IN bacteria

For DNA extraction, 2 g of soil was mixed into a thick slurry with deionized water (18.2 MΩ and 0.2 µm filtered) and 50 µL transferred to the DNA extraction tube. For the lodgepole pine forest litter, 2 g was mixed with 40 mL deionized water, shaken for 20 min at 250 rpm, then 0.5 mL transferred to a 1.5 mL microfuge tube, spun at 22,500 g for 5 min and all but 50 µL of the supernatant removed. This was then transferred to the DNA extraction tube using solution MD1 of the DNA extraction kit. DNA extraction was performed using the recommended protocol of the PowerLyzer™ UltraClean® Microbial

DNA Isolation Kit (MO BIO Laboratories Inc.), with homogenization on a FastPrep® bead beater (BIO 101 Inc.) at setting 4 for 5 min.

Quantitative PCR of the *ina* gene was initially attempted using the method described in Hill et al. (2014). However, a lack of detectable amplicons in any samples combined with the high diversity of soil genomic DNA led to significant mis-priming,

which limited the method's sensitivity. We therefore spiked standard *ina* gene PCRs with a range of known *ina* gene copies and used detection of a discernible amplicon on gels to gauge the approximate limits of detection.





Each PCR contained 1 U of GoTaq® Hot Start Polymerase (Promega) in 1× Colorless GoTaq® Flexi Buffer (no Mg in buffer), 0.9 μM forward primer 3308f (5′ GGCGATMGVAGCAAACTSAC 3′), 0.9 μM reverse primer 3462r1 (5′ TGTAVCKTTTSCCGTCCCAG 3′), 0.2 mM each dNTP, 1.25 mM MgCl$_2$, 4% DMSO, 2 μL of DNA extract, 0.5 μL *ina* gene spike and deionized water to a total volume of 25 μL. Spikes ranged from 60 to 1185 *ina* gene copies using genomic DNA extracted from a pure culture of *P. syringae* Cit7 (see Hill et al. (2014) for details of the isolate, culturing and preparation of standards).

Cycling conditions were an initial denaturation at 95 °C for 2 min, followed by 40 cycles of 94 °C for 15 s and 54 °C for 25 s (combined primer annealing and extension). Amplification was performed on a Bio-Rad DNAEngine® (Hercules). After amplification, products (194 bp) were electrophoresed in 1.5% MetaPhor® agarose gels (Cambrex) in 1× sodium borate buffer at 200 V for 35 min, using ethidium bromide for visualization. A 50 bp ladder (G4521, Promega) was used for sizing. Presence of an unambiguous amplicon on gels was used to estimate the limit of detection of *ina* genes for each soil's DNA, and converted to theoretical upper limits of IN bacteria per gram dry soil (there is one copy of the *ina* gene per bacterium).

**2.9 Direct isolation of INPs from soil**

Freezing triggered by an INP can be exploited as a means to isolate it. Initially, a $10^{-5}$ dilution of sagebrush soil was prepared in 0.2 μm pore diameter filtered deionized water. Then, 50 μL aliquots were dispensed into a 96 well PCR tray and the tray cooled to -7.0 °C. Frozen wells were noted and the plate thawed. The contents of a well that froze was transferred to a sterile microplate lid (Nunc, Thermo Scientific) and divided into an array of 1 μL droplets. The plate was covered, placed on a cold block at ≈-15 °C for 10 min, uncovered and using a stereo microscope the single frozen droplet noted. The plate was warmed, 2-4 μL of deionized water added to the frozen droplet, and the droplet then divided into an array of 0.1 or 0.2 μL droplets for freeze testing. This cycle was repeated 2-3 further times. The lid was then transferred to a microscope, the droplet containing the INP observed as it evaporated, and the residual photographed.

**2.10 Ice nucleation by sagebrush tissues**

The semi-arid sagebrush shrubland soil had a particularly pronounced onset of ice nucleation activity, increasing from ≈30 to almost a million INPs g$^{-1}$ soil between -5 and -6 °C. This suggests a single class of INP was responsible. To test whether sagebrush tissues were the INPs responsible, shoots (leaves and small stems) and roots up to 1 mm diameter were collected at the sagebrush shrubland site in February 2013 and tested.

Foliage and roots were washed to remove loose soil, placed in fresh plastic bags filled with deionized water and sonicated for several minutes to further dislodge soil particles. The water was replaced and the process repeated five times for a total of 30 min of sonication. A subsample was then ground under liquid nitrogen in a pestle and mortar (cleaned by soaking in 5%





$H_2O_2$ for several hours followed by repeated rinses in deionized water). The ground root powder was examined under a microscope and contained root cell fragments plus bacteria and occasional pieces of fungal hyphae. Serial dilutions in deionized water were tested for INP content. A sample of the initial 50-fold dilution was re-tested after heating to 105 °C for 15 min to determine INP heat sensitivity.

## 3 Results and discussion

All soils presented contained abundant INPs active at warm temperatures (Fig. 1). While the onset of nucleation ranged over 2.5 °C, all soils demonstrated a pronounced increase below -5 °C, while below approximately -8 °C the rate of increase lessened to become progressively log-linear. INP concentrations at -10 °C ranged from ≈$10^6$ to ≈$5 \times 10^7$ $g^{-1}$ dry soil, with levels in the native grassland and agricultural soils comparable to those found in grassland and agricultural topsoils in Germany, Hungary and England (Conen et al., 2011; O'Sullivan et al., 2014, soils A, C and D). INPs in the two pine forest litters were, however, lower than reported for comparable USA High Plains litters by Schnell and Vali (1976); they noted they were similar to values of ≈$10^9$ INPs $g^{-1}$ fresh litter found in D-type vegetation. Unexpectedly, the lowest concentrations were found in ponderosa pine forest litter, even though the organic matter content of this sample was high (≈20%, Table 1). INP production by soils and litters is thus not only a function of climatic region (Schnell and Vali, 1976) and SOM content (Conen et al., 2011), but is also dependent upon the resource quality of the SOM supplied by the vegetation, among other factors.

### 3.1 INPs in the total soil organic matter

Reductions in INPs caused by SOM removal with $H_2O_2$ (Fig. 2) were similar to or greater than found by Conen et al. (2011) and greater than found by O'Sullivan et al. (2014) who used a gentler peroxide treatment. At the lower temperature limit (-15 to -18 °C), INPs were reduced by 2-3 orders of magnitude, comparable to the findings of Conen et al. (soils B-D; 2011), but more than a 10-fold greater impact than observed by O'Sullivan et al. (2014). By contrast, no effect of $H_2O_2$ digestion was observed for kaolinite, in agreement with the results of Tobo et al. (2014), and as also found for the minerals montmorillonite and K-feldspar (Conen et al., 2011; O'Sullivan et al., 2014).

There was good agreement between results obtained here, using immersion freezing on bulk soil suspensions, with INP spectra below -18 °C obtained from aerosolized soil dusts (0.6 μm diameter) of the sugar beet and organic alfalfa fallow soils, and measured using the continuous flow diffusion chamber (Tobo et al., 2014) (Fig. 2), suggesting that organic forms would also comprise the bulk of INPs active at warmer temperatures in dusts raised from these agricultural soils.



## 3.2 INPs in fulvic and humic acids

Soil organic matter comprises a spectrum of forms ranging from fresh plant inputs to ancient, refractory organic matter. It is often divided into the broad classes of decomposable plant material, resistant plant material, microbial biomass, humus and inert organic matter (Coleman and Jenkinson, 1999). The humus fraction, a complex mixture of biologically transformed
plant and microbial debris, usually predominates, and has itself been classically sub-divided into three groups: fulvic acids, humic acids and humin. Fulvic acids are the fraction that dissolve in 0.1 M HCl, humic acids the fraction that subsequently solubilize in 0.1 M NaOH, and humin the residue (Powlson et al., 2013). These three groups delineate broad and "operationally-defined" — as opposed to "ecologically-defined" — pools of humic substances (Wander, 2004).

Humic and fulvic acids are of particular interest because of their apparent similarity to atmospheric HULIS (HUmic-LIke Substances), the often significant constituent in water-soluble organic carbon extracted from aerosols (Graber and Rudich, 2006; Wang and Knopf, 2011). However, Graber and Rudich (2006) concluded that HULIS are both smaller (MW <1,000) and less aromatic than terrestrial humic substances, and more likely to be the product of the oligomerization of smaller molecules while airborne or the breakdown and reassembly of fulvic and humic acids lofted from soil surfaces. HULIS thus
appears to be similar to, but not the same as, humus. Studies using commercially available fulvic and humic acids as surrogates for HULIS found they possessed unremarkable ice nucleation activities (Kanji et al., 2008; O'Sullivan et al., 2014; Wang and Knopf 2011). O'Sullivan et al. (2014) recorded the warmest activity with sieved suspensions: onset of activity for Suwanhee River fulvic acid was ≈-14 °C and for Leonardite humic acid it was ≈-19 °C. Fornea et al. (2009) measured onset of freezing temperatures of -10.5 °C for Pahokee peat, but they used individual particles of the intact soil. It
should be noted that the standard extraction method used would also solubilize and remove most atmospheric HULIS forms from the SOM (Feczko et al., 2007).

Removal of fulvic and humic acids from the pasture soil only reduced INPs active ≥-7 °C (Fig. 3), and these may have been biological macromolecular INPs denatured by the pH 13 extractant. The small molecular weights and the diversity of forms
of fulvic and humic acids make it improbable that they posses significant ice nucleation activity. By contrast, the residual humin is compromised in large part of macromolecules or aggregates of peptides, aliphatics, carbohydrates, lignin and the bacterial cell wall polymer peptidoglycan (Simpson et al., 2007), and hence is a more likely source of IN molecules.

## 3.3 INPs from sterols and monolayers of aliphatic alcohols

Several classes of naturally occurring organic compounds are effective INPs (Bashkirova and Krasikov, 1957; Head, 1961; 1962; Fukuta, 1966). Crystals of some sterols, for example, trigger nucleation >-5 °C, although the fungal sterol ergosterol and the plant sterol β-sitosterol were found to be inactive (Head, 1962; Fukuta and Mason, 1963).





Under optimal conditions, monolayers of aliphatic alcohols can also induce ice nucleation at temperatures ranging from ≈-14 °C for $C_{14}H_{29}OH$ to just below -1 °C for $C_{31}H_{63}OH$ (Gavish et al., 1990; Popovitz et al., 1994; see also Rosinski, 1980). Nucleation is facilitated by alcohols self-assembling into two-dimensional crystalline clusters on air:liquid or liquid:liquid

interfaces, with their hydroxyl groups embedded into the water surface and arrayed with a spacing closely matching hexagonal ice (Gavish et al., 1990; Popovitz-Biro et al., 1991). Intriguingly, Schnell and Vali (1972) showed that volatiles released from dry leaf litters heated to ≈100 °C and then condensed onto a clean surface were highly ice nucleation active. However, in the two soils tested here, the removal of organics with chloroform had no significant effect upon the INP spectra (Fig. 4), suggesting that IN organic crystals or long-chain alcohols were not present, or did not occur in sufficient quantities

or the required purity.

### 3.4 Heat-labile INPs

As shown in Fig. 5, the 60 °C treatment had essentially no impact upon the soils' INP activation temperature spectra, apart from some reduction ≥-6 °C in the sugar beet and lodgepole pine soils. This general insensitivity suggests that labile IN

proteins or protein complexes from IN bacteria or other microbial or plant sources were not present, or not significant contributors, of high-temperature INPs to the soils.

Heating suspensions of IN bacteria, isolated from crops at the agricultural research centre, to 105 °C denatured and deactivated their Ina proteins (Hill et al., 2014). Likewise, 98 °C treatment almost eliminated the ice nucleation activity of all

but a couple of local isolates of IN *M. alpina* (Fröhlich-Nowoisky et al., 2015), while 90 °C heating lowered the median activity of IN *F. avenaceum* by 10 °C. Correspondingly, the 105 °C treatment reduced the INP concentration active at -9 °C in all soils by ≥10-fold. In the lodgepole pine litter this reduction extended to at least -15 °C. Interestingly, for boundary layer aerosols collected above crops in northern Colorado (Garcia et al., 2012), the reduction in INPs following 98 °C heating was similar to that found in these soils but, like with the lodgepole pine litter, extended further, to ≈-18 °C (the limit

of the measures in that study). In three out of the four soils tested by Conen et al. (2011) heating to 100 °C reduced INPs active at -12 °C by 70-98% while in three of the four agricultural soils tested by O'Sullivan et al. (2011) a 90 °C treatment reduced INPs active at -10 to -12 °C by around 10-fold.

### 3.5 IN bacteria

Lysozyme had a minimal overall impact upon INP concentrations, but did reduce INPs active at ≥-6 or ≥-7.5 °C, apart from in the lodgepole pine soil (Fig. 6). In this respect – its affect on high-temperature-active INPs – it generally corresponds to





the findings of Christner et al. (2008), of lysozyme-sensitive material accounting for 50% and 25% of INPs active ≥-9 °C in precipitation samples from Montana and Louisiana, respectively. The bacteria deactivated by the lysozyme may have included the known IN bacteria at levels below the detection limits of the PCR test (see below) and/or other IN bacterial species.

Evidence exists both for and against the occurrence of IN bacteria (species of *Pseudomonas*, *Pantoea* and *Xanthomonas*) as natural components of the soil microflora. They have been isolated at ≈$10^5$ g$^{-1}$ from surface field soil (Lindemann at el., 1982), and IN *Ps. borealis* was reported as a natural component of Arctic tundra topsoil (Wilson et al., 2006). Garcia et al. (2012) also found ≈$10^5$ *ina* genes g$^{-1}$ in the surface soil of a just-harvested corn field, but it is possible these came from pulverized corn tissue dusting the surface. Vali et al. (1976) also demonstrated the ability of a locally isolated *Ps. syringae* to colonize leaf litter. By contrast, Conen et al. (2011) predicted that the contribution of IN bacteria to soil INPs would be negligible based on previous studies showing that *Ps. syringae* typically did not survive more than a few weeks after addition to soils (eg, McCarter et al., 1983; Goodnow et al., 1990).

Direct testing for the presence of IN bacteria, using PCR with primers designed to amplify most alleles of the *ina* gene (Hill et al., 2014), which codes for the ice-active Ina protein, did not detect them in any of the soil DNA extracts (Fig. 7a). When soil DNA was spiked with DNA from IN *Ps. syringae* Cit7, the *ina* genes were readily amplified, although with somewhat limited sensitivity due to mis-priming and co-amplification of non-target DNA (Fig. 7b). Using spiking tests with a range of *ina* gene copies, the lower detection limits were determined. When these were expressed as a fraction of INPs, the IN bacteria accounted for 0.5 to 4% of INPs active at -12 °C in the pasture and organic grass/alfalfa soils, respectively (-12 °C is the temperature at which a single Ina protein nucleates (Govindarajan and Lindow, 1988)). Under optimal conditions, the proportion of IN bacteria with ice nucleation activity at -10 °C can be 10% or higher (Kim et al., 1987; Nemecek-Marshall et al., 1993; Attard et al., 2012). Even if the populations possessed a 10% frequency of nucleation, the IN bacteria would then only account for <0.5% of INPs at -12 °C.

It is possible that Ina proteins could accumulate in soil by being adsorbed onto clay and/or organic matter (Schnell, 1977), thereby being protected from enzymatic digestion. Accordingly, even a very small IN bacterial population could accumulate to exert a disproportionate influence upon the INP content of soils. This mechanism could also apply to other cell-free INPs.

## 3.6 IN fungi

Screening of fungi isolated from agricultural and natural ecotype topsoils in Wyoming (including several of the soils used in this study) found a single species of IN fungus, *Mortierella alpina*, to be widespread and also abundant in soils with fresh



inputs of plant material (Fröhlich-Nowoisky et al., 2015). *Mortierella alpina* is a saprobe utilizing decaying organic matter (Wagner et al., 2013), and due to its ability to solubilize phosphorus also develops a mutualistic association with another fungal genus that forms mycorrhizal symbioses with sagebrush (Wicklow-Howard, 1994; Zhang et al., 2011). All 39 isolates tested by Fröhlich-Nowoisky et al. (2015) initiated freezing at -5 to -6 °C. They also typically released $10^8$ to $10^9$ small (<10 nm) INPs per gram fresh weight of mycelium that nucleated between -5 and -8 °C. For the three most common clades, which occurred in agricultural and lodgepole pine forest soils, digestion of isolates with papain lowered the INP concentration by ≈3-4 orders of magnitude (Fröhlich-Nowoisky et al., 2015), suggesting the INP was a protein.

In this study, digestion with papain lowered INPs active ≥-11 °C in the organic grass/alfalfa, the pasture and the sagebrush soils by ≥75% (Fig. 8). Indeed, in these soils its effect was comparable to that of 105 °C heating. By contrast, papain only deactivated INPs active ≥-7 °C in the sugar beet soil and had no impact on lodgepole pine forest soil, even though IN *M. alpina* was readily isolated from both (Fröhlich-Nowoisky et al., 2015). This may have been because these soil samples were taken at different times from those in the Fröhlich-Nowoisky et al. (2015) study; because many fungal species have very sporadic occurrences of vegetative growth before dying away, leaving their spores behind, a high relative abundance with dilution plating does not automatically equate to a significant biomass (Warcup, 1957). Alternatively, the papain-susceptible INPs in the other three soils were not from *M. alpina*, but from another source.

Several species of fungi from the genus *Fusarium* are also known to be IN (Pouleur et al., 1992; Richard et al., 1996). These species contain strains that are plant pathogens as well as saprophytes, and are promoted by agricultural practices. Digestion of cultures of *F. avenaceum* and *F. acuminatum* with Proteinase K significantly reduced ice nucleation activity, suggesting that a protein is responsible (Hasegawa et al., 1994; Humphreys et al., 2001). In the agricultural soils tested, incubation with Proteinase K had a minimal affect upon overall INP concentrations. In the organic grass/alfalfa and sugarbeet soils it reduced INPs active ≥-6 °C by ≈10-fold (Fig. 9), as would be expected for deactivation of IN *Fusarium* (Hasegawa et al., 1994; Richard et al., 1996; Pouleur et al., 1992; Humphreys et al., 2001) and *Mortierella*, but it had no effect on the roadside pasture soil. The cultivated soils thus contained some Proteinase K-sensitive IN microflora.

### 3.7 Direct isolation of INPs from sagebrush soil

We present the first images of INPs directly isolated from soil. INPs active at -7 °C in the sagebrush shrubland soil were isolated by repeated cycles cooling of arrays of droplets to identify the INP-containing droplet, addition of de-ionized water to dilute that droplet, and its further subdivision and subsequent freeze testing. Starting with a $10^{-5}$ dilution of the soil suspension, a further 5-6 cycles of freezing and droplet subdivision were required to obtain a single visible particle in the evaporated droplet residue.



Two INPs were isolated. INP1 was roughly 20 × 5 µm and appeared to be a complex entity (Fig 10a). After isolation and photography, INP1 was re-suspended in 1 µl water, which was divided into 4 droplets and these freeze tested. This time two wells froze, and in all four droplets there were numerous fragments of the original piece (Fig. 10b). In sagebrush surface soils, freeze/thaw cycles occur frequently from mid-Autumn to mid-Spring, providing multiple opportunities for such fracturing and multiplication of INPs. INP2 appeared to be a roughly 30 × 20 µm piece of gel (Fig. 10c&d). It was partly squashed by capillary forces when mounted between slide and coverslip (Fig. 10d) and was semi-transparent. Although the matrix may have been mucilage secreted by roots (Foster et al., 1983), exopolysaccharide and alginate gels are also produced by microbes, and the two often intermix forming mucigels (Foster, 1986). The actual INP was potentially one of many things: either a component of the gel itself or one of it many inclusions.

Filtration tests of a sagebrush soil suspension showed that 68% of INPs active at -7 °C were >5 µm diameter and 88% >0.45 µm. Hence, the randomly selected INPs isolated here were also large. It is possible that the true INP was an accompanying component too small to be visible. However, the fragmentation of INP1 after six freeze/thaw cycles leading to the production of two INPs suggests that the large particle was indeed the source.

### 3.8 Ice nucleation by sagebrush tissues

The particularly pronounced onset of ice nucleation activity in sagebrush shrubland soil (Fig. 2) suggested a single class of IN material was responsible. Since Wyoming big sagebrush (*Artemisia tridentata*) contributes organic matter directly to the soil via leaf fall and the shedding of fine roots and root hairs, its tissues may be a direct source of these INPs. Indeed, powdered sagebrush shoots and roots contained a distinct and abundant class of INPs that nucleated at ≈-12 °C (Fig. 11). The relatively warm temperature of activity of sagebrush tissue INPs suggests it may play a functional and protective role for the plant, although its onset temperature is much lower than that possessed by IN peach wood tissue (≈-2 °C, Gross et al., 1988). Interestingly, *A. tridenta* seedlings from comparable altitudes exhibited significant damage (>50% loss of potential function of Photosystem II) when chilled to between -12.5 and -15 °C (Loik and Redar, 2003), suggesting that sagebrush tissues may naturally supercool until being nucleated by these endogenous INPs. Heating to 105 °C completely deactivated the INPs active at ≈-12 °C. Likewise, they appeared to be lost upon decomposition, since there was no evidence of an increase in INPs active around this temperature in the soil (Fig. 2). Cellulose may also have contributed to the residual INP activity of the sagebrush tissue that remained after heating (Hiranuma et al., 2015).

In addition to the highly active entity at ≈-12 °C, sagebrush roots also had a lesser source of INPs with a profile similar to that of the soil (Fig. 11b). This high-temperature source may have been contributed by the diverse microflora that colonizes





roots, from soil biological material still attached to the epidermis of the roots after washing, or from arbuscular mycorrhizal fungi and other microflora inhabiting the root cortex (Watson, 1987). Notably, one of the few fungal genera that form a mycorrhizal symbiosis with sagebrush also develops a mutualistic association with *M. alpina* (Wicklow-Howard, 1994; Zhang et al., 2011).

## 4 Conclusions

Ice nucleating particles denatured by heating (105 °C) accounted for most INPs active >-10 °C. A range of entities both within the soil microflora and SOM will be heat-labile. Proteins will be denatured; molecular complexes, organelles and structures (eg, ribosomes, flagella, membranes) entirely disrupted; crystals dissolved; and aggregates of storage polymers (eg, polyhydroxyalkanoates) and lipid bodies dispersed. Immersion in hot water could also affect the IN ability of minerals. For example, the IN activity of K-feldspar (Atkinson et al., 2013) may be altered by a change in composition of surface ions (Zolles et al., 2015). However, the lack of any effect of heating samples to 60 °C suggests this wasn't the case.

While lysozyme and Proteinase K had minimal overall impacts upon INP concentrations, papain protease reduced INP numbers significantly in two agricultural soils and in the sagebrush shrubland soil, suggesting that in these a protein was responsible. This may have been due to the presence of *M. alpina* (Fröhlich-Nowoisky et al., 2015), but equally may have been IN proteins from other microbes and/or the soil fauna (eg, protozoa, mites, nematodes).

Two IN particles, isolated by exploiting their IN activity at -7 °C, were both complex entities. Freeze fracturing of one into at least two INPs demonstrated that repeated freeze-thaw events can cause subdivision and, hence, multiplication of INPs.

Fig. 12 summarizes the sources of INPs in the pasture soil. With variations in proportions it can be used as a general summary of likely INP composition in arable and grassland topsoils. Notable is the segment of organic INPs active below -10 to -12 °C that was unaffected by any challenge short of oxidation with $H_2O_2$. Their refractoriness can be used to rule out some potential sources:

- they are not proteins or peptides, due to their resistance to 105 °C heating and insensitivity to papain and Proteinase K (they could be proteins protected by adsorption onto clay, but this would not explain their presence in the lodgepole pine litter);
- they are not peptidoglycan from bacteria since they survived lysozyme digestion;





- they are unlikely to be chitin and related fungal cell wall polysaccharides since Pummer et al. (2013) found no appreciable ice nucleation activity above -26 °C in fungal spores from a range of common fungi (other than in a known IN *Fusarium*);
- they are not IN aliphatic monolayers, organic IN crystals or even membranes, since steeping in chloroform had no effect;
- and they do not appear to be fulvic or humic acids.

Hence, they are some component/s of plant material, microbial biomass and/or humin. Humin is compromised in large part of macromolecules or aggregates of peptides, aliphatics, peptidoglycan, carbohydrates and lignin. Ruling out the first three leaves lignin or carbohydrates as possibilities. Lignin seems an unlikely candidate as an INP due to its disordered structure and relative hydrophobicity. Cellulose, a carbohydrate, may account for some of the activity colder than -15 °C (Hiranuma et al., 2015).

The variable effectiveness of the challenge tests in different soils indicates that soil INPs come from a range of sources, occur with differing relative abundances, and may be protected by different mechanisms. For example, the complete insensitivity of INPs in the lodgepole pine forest litter to any tests other than heat and $H_2O_2$ digestion suggests that this population differs from those in non-forest soils.

Since SOM comprises a diverse continuum of forms, ranging from fresh plant inputs to ancient inert organic matter, it should be expected that both the sources and their intrinsic temperatures of nucleation will be similarly broad. This can be inferred from the smooth curves of the INP temperature spectra, each being the sum of many underlying activity distributions. Organic INPs will also occur as organo-mineral complexes, which will presumably affect their activity and protect them by spatial isolation and adsorption onto clays (Schnell, 1977; Kögel-Knabner et al., 2008).

All soils, with the relative exception of ponderosa pine litter, were rich potential sources of organic, warm-temperature-active IN aerosols. Agricultural intensification in semi-arid regions of the western U.S. over the past 200 years has led to a more than four-fold increase in atmospheric dustiness (Neff et al., 2008). The increased wind erosion also selectively removes the fine and light fractions, which are enriched in particles <10 μm in diameter and organic matter approximately six-fold (Van Pelt and Zobeck, 2007). Biological particles, predominantly bacteria and fungi, accounted for an average of 40% of the organic carbon in particles <10 μm at Storm Peak Laboratory in western Colorado (Wiedinmyer et al., 2009) and were present in dust events intersecting the site (Hallar et al., 2011). INP emissions by these western U.S. landscapes will thus be influenced by soils that are more prone to destabilization by both previous and current human activities.



## 5 Acknowledgments

We thank Jenna Meeks and Bob Baumgartner for access to the soils at SAREC, William Stump for advice concerning experimental farm soils, and Prakriti Bista for elemental analyses. This work was funded by National Science Foundation grants 1358495 (PD and TH), 0841542 (TH and GF) and 0841602 (PD). J. Fröhlich-Nowoisky acknowledges the Max Planck Society and the Ice Nuclei research Unit of the German Research Foundation (DFG FR3641/1-2, FOR 1525 INUIT) for project funding. Y. Tobo acknowledges the Japan Society for the Promotion of Science (JSPS) Postdoctoral Fellowships for Research Abroad.

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

## 7 Tables and figures

Table 1. Descriptions of sites sampled.

| Site | Sampling date(s) | Lat. | Long. | Vegetation | Soil type | C$_{org}$ (%) | N (%) | pH |
|---|---|---|---|---|---|---|---|---|
| | | | | **Agricultural ecotypes** | | | | |
| Grass/alfalfa fallow [a] | 2 Mar 2011 & 17 May 2011 | 42.12266 | -104.38585 | Dead material on surface from previous year's sowing of alfalfa, orchard grass, and meadow brome. Organically managed field. | Haverson & McCook light brownish-gray floodplain loams.[b] | 0.55 | 0.076 | 8.1 |
| Sugar beet[a] | " | 42.12878 | -104.39516 | Bare at sampling. Previous year was Roundup Ready sugar beet. | " | 0.75 | 0.11 | 8.15 |
| Pasture[a] | " | 42.13243 | -104.39428 | Roadside, low-input pasture of smooth and downy bromes. | " | 2.7 | 0.46 | 7.85 |
| | | | | **Native ecotypes** | | | | |
| Native Grassland | 24 May 2011 | 41.2881 | -106.11124 | Bluebunch wheatgrass, Idaho fescue, western wheatgrass, and threetip sagebrush. | Greyback very cobbly sandy loam; outwash from alluvial fan. Surface layer grayish brown to brown very cobbly sandy loam.[c] | 2.2 | 0.27 | 6.45 |
| Sagebrush shrubland | 2 Dec 2012 & 17 Feb 2013 | 41.03159 | -106.00170 | Semi-arid rolling Wyoming big sagebrush shrubland. | Fine, loose sand with some surface cobbles. | 1.5[e] | - | 7.15 |
| Lodgepole pine forest | 24 May 2011 & 2 Dec 2012 | 41.32439 | -106.16062 | Lodgepole pine, with understory of elk sedge, low sedge, creeping juniper, Oregon grape, kinnikinnick, woods rose, and heartleaf arnica. | Ansile-Granile gravelly sandy loam. 5 cm layer of needles and bark residue.[c] | 31[e] | 2.0 | 5.9 |
| Ponderosa pine forest | 22 July 2012 | 39.10330 | -105.10395 | Dominated by ponderosa pine, with sparse grasses and sedge ground cover. | Gravelly alluvium, classified as loamy, mixed Eutroboralfs or Aridic Haploborolls.[d] | 11.6 | 0.6 | 5.05 |

[a] At the University of Wyoming's Sustainable Agricultural Research and Extension Center (SAREC), near Lingle, Wyoming.

[b] Soil survey of Goshen County, south part, Wyoming, 1971. United States Department of Agriculture, Soil Conservation Service, 102 pp.

[c] Soil survey of Albany County Area, Wyoming, 1998. United States Department of Agriculture, Natural Resources Conservation Service, U.S Government Printing Office, 540 pp.

[d] Moore, R. 1992. Soil survey of Pike National Forest, eastern part, Colorado, parts of Douglas, El Paso, Jefferson, and Teller counties. U.S. Gov. Print. Office, Washington, DC.

[e] Soil organic carbon obtained by loss-on-ignition at 550 °C for 3 h, and then multiplying values by 0.58 (De Vos et al.,

2005).




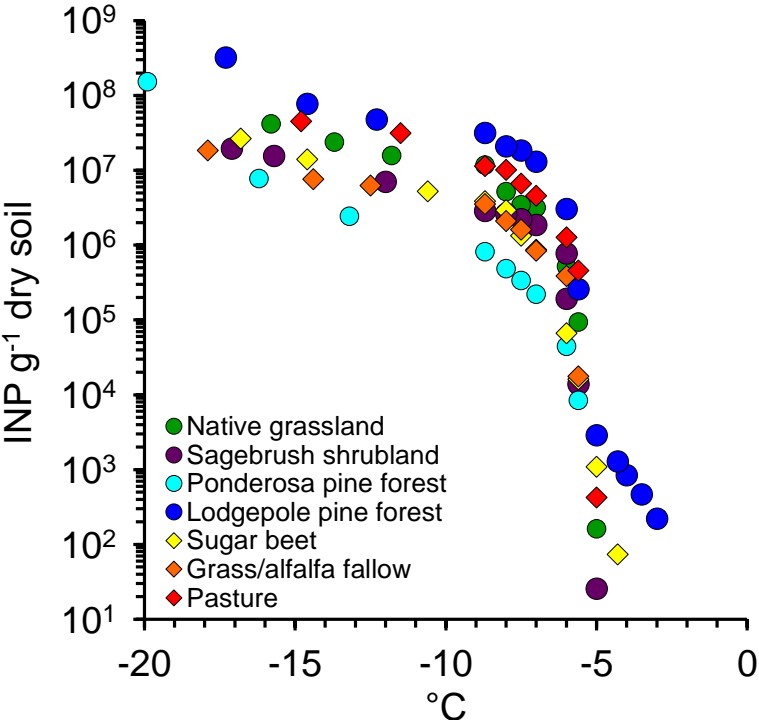

Fig. 1. Ice nucleating particle concentrations in four natural ecotype and three agricultural soils.





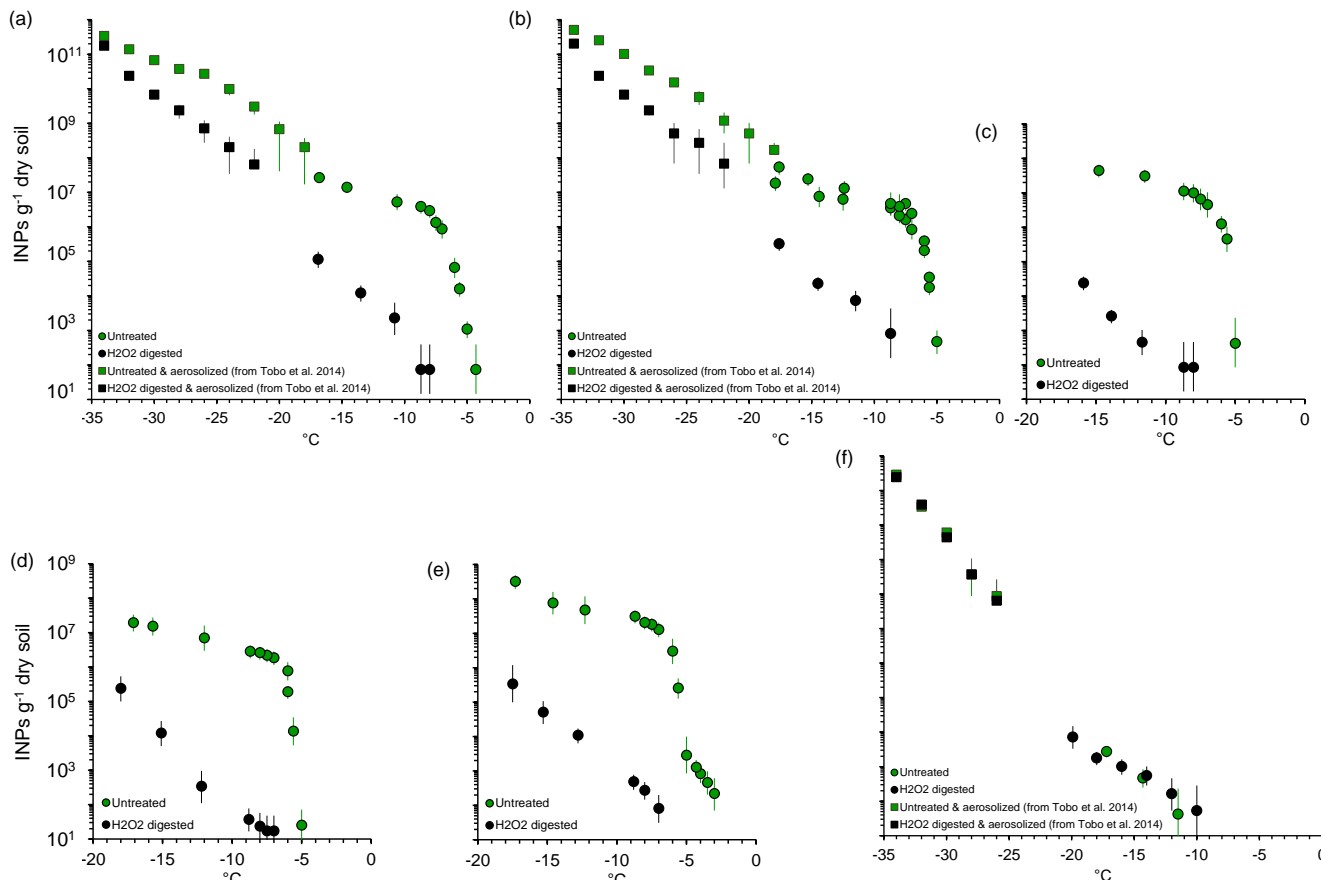

Fig. 2. Effect of removal of SOM by $H_2O_2$ digestion upon INP concentrations of soils. Soils are (a) sugar beet, (b) organic grass/alfalfa fallow, (c) pasture, (d) sagebrush shrubland, (e) lodgepole pine forest, and (f) kaolinite. The sugar beet, organic grass/alfalfa and kaolinite figures include data of aerosolized soil dusts (0.6 μm dia.) from Tobo et al., (2014) measured using a continuous flow diffusion chamber.



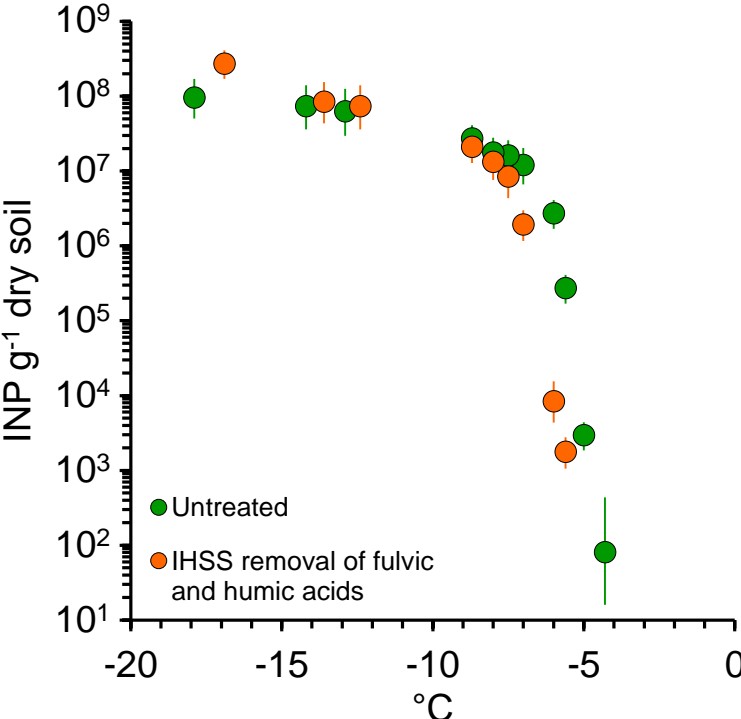

Fig. 3. Effect of removal of fulvic and humic acids upon INP content of pasture soil.



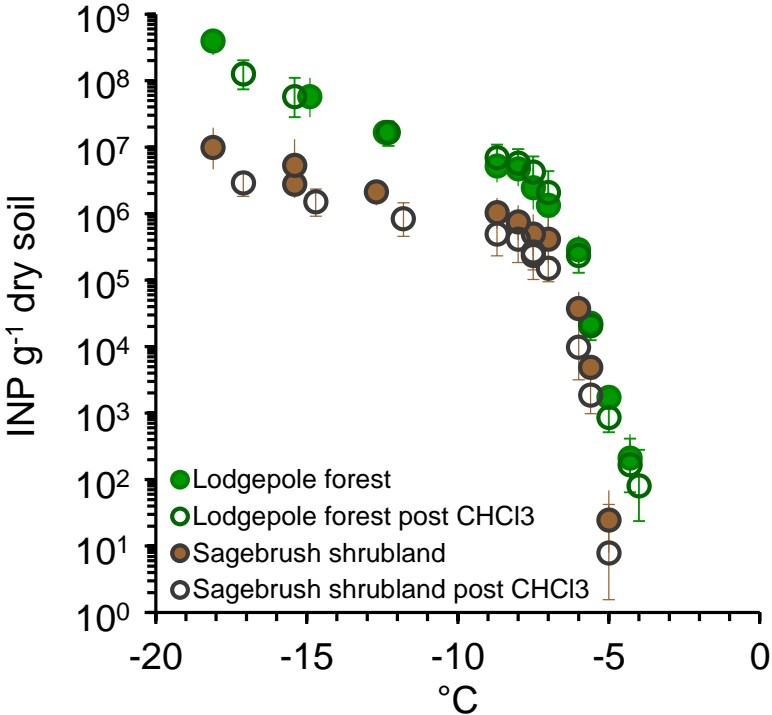

Fig. 4. Effect of removal of organics, such as sterols and aliphatic alcohols, with chloroform upon INP temperature spectra of forest and shrubland soils.





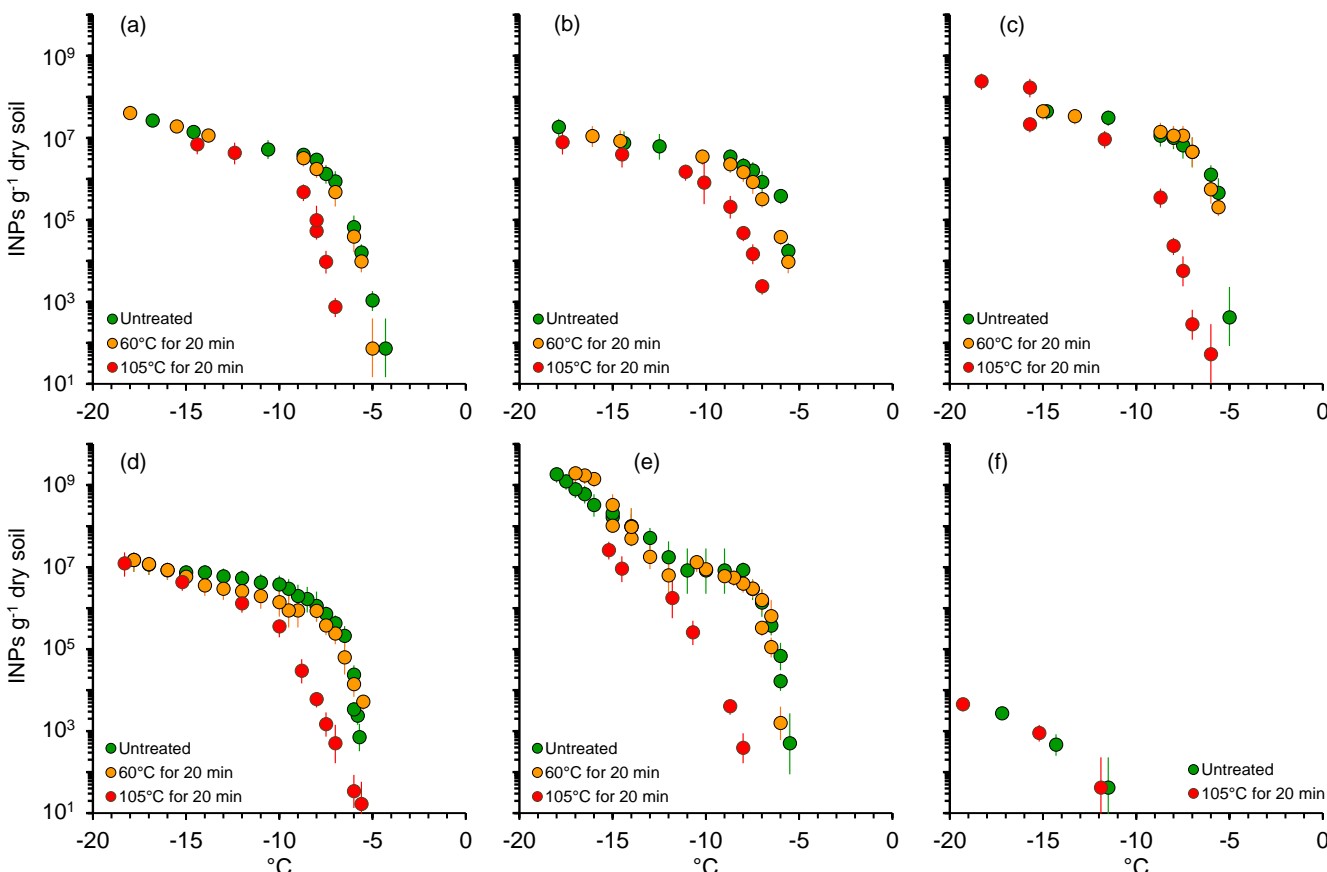

Fig. 5. Effect of heating in water to 105 °C upon INPs in soils. Soils are (a) sugar beet, (b) organic grass/alfalfa fallow, (c) pasture, (d) sagebrush shrubland, (e) lodgepole pine forest, and (f) kaolinite.



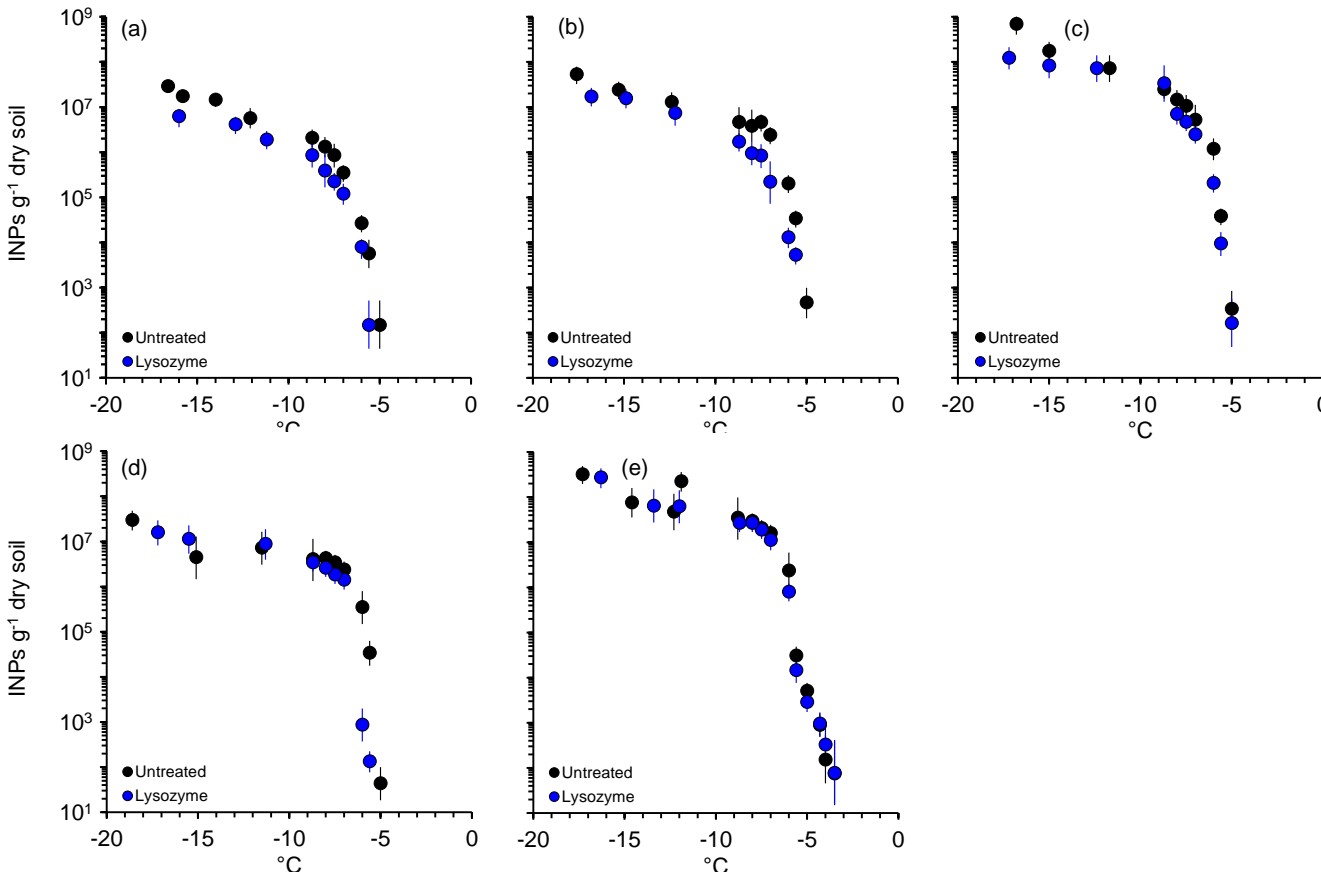

Fig. 6. Effect of lysozyme digestion upon soil INPs. Soils are (a) sugar beet, (b) organic grass/alfalfa fallow, (c) pasture, (d) sagebrush shrubland and (e) lodgepole pine forest.



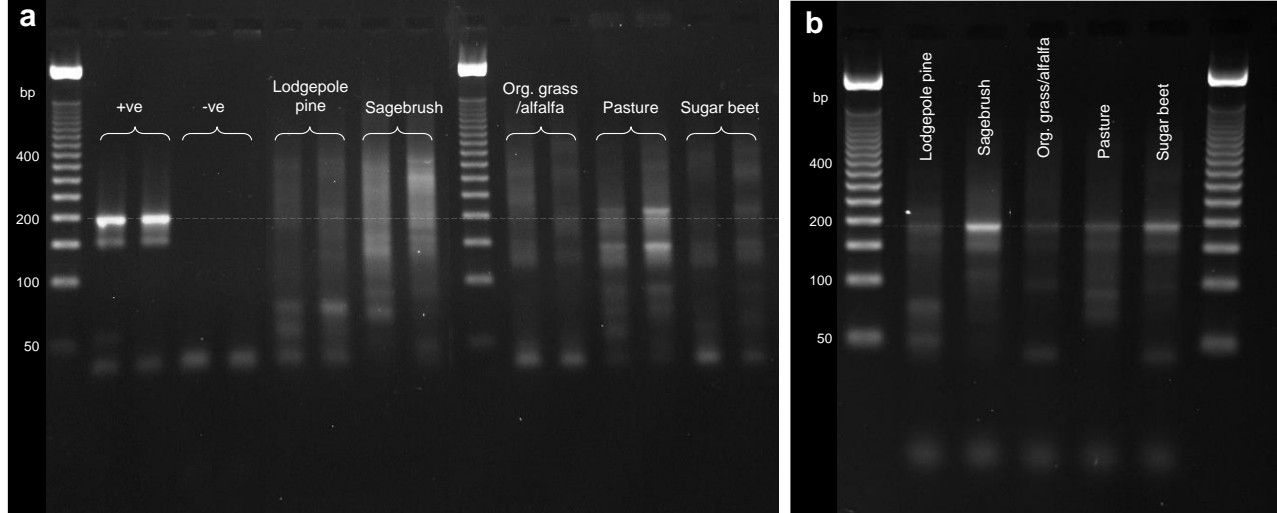

Fig. 7. Test for *ina* gene presence in soil DNA. (a) No detectable *ina* gene amplicons (194 bp, as indicated by the faint dashed line) were discernable in any of the soils. (b) Soil DNA spiked with 60 *ina* gene copies. Amplicons were visible in all samples, although differing in intensity due to sub-optimal amplification in most.




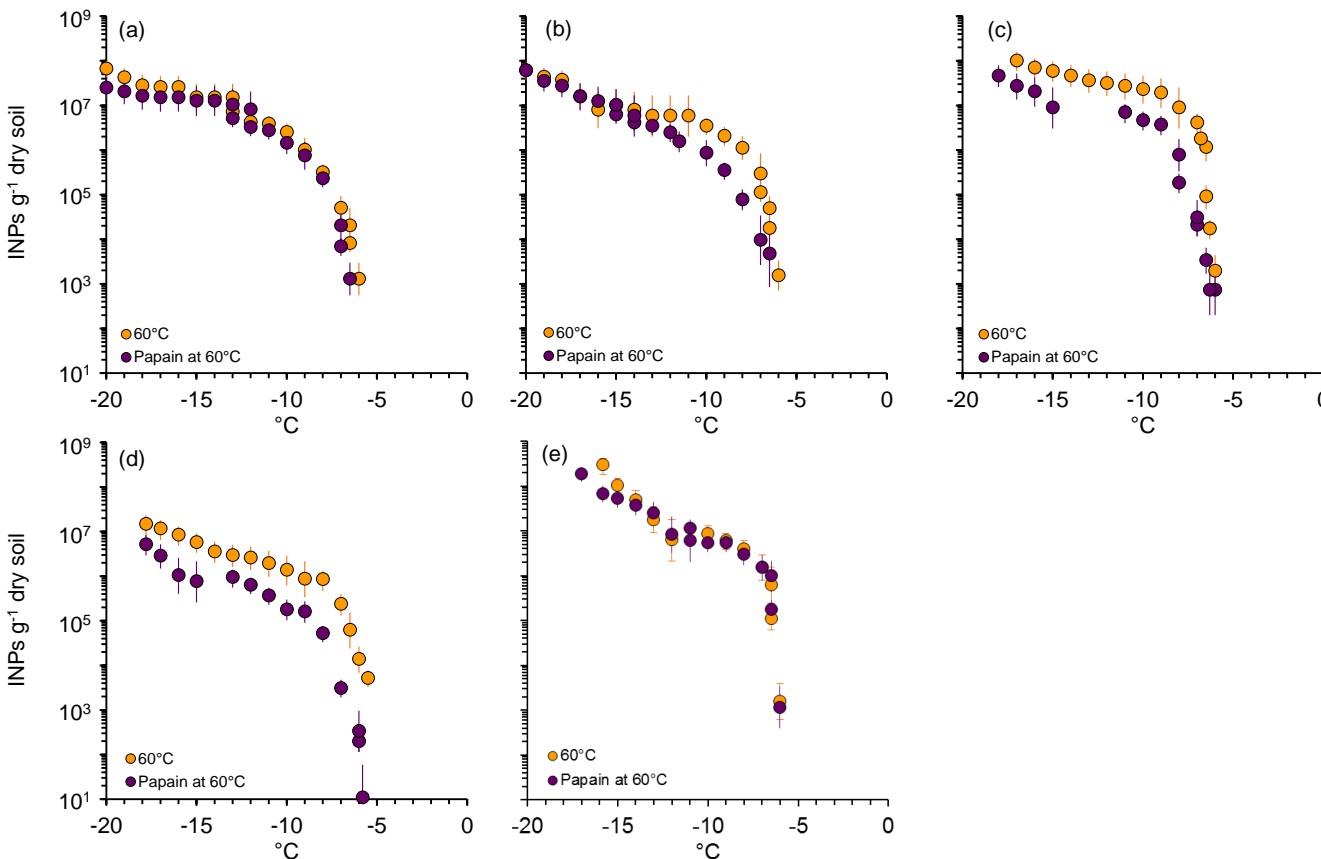

Fig. 8. Effect of papain protease digestion upon INPs in soils. Soils are (a) sugar beet, (b) organic grass/alfalfa fallow, (c) pasture, (d) sagebrush shrubland and (e) lodgepole pine forest.



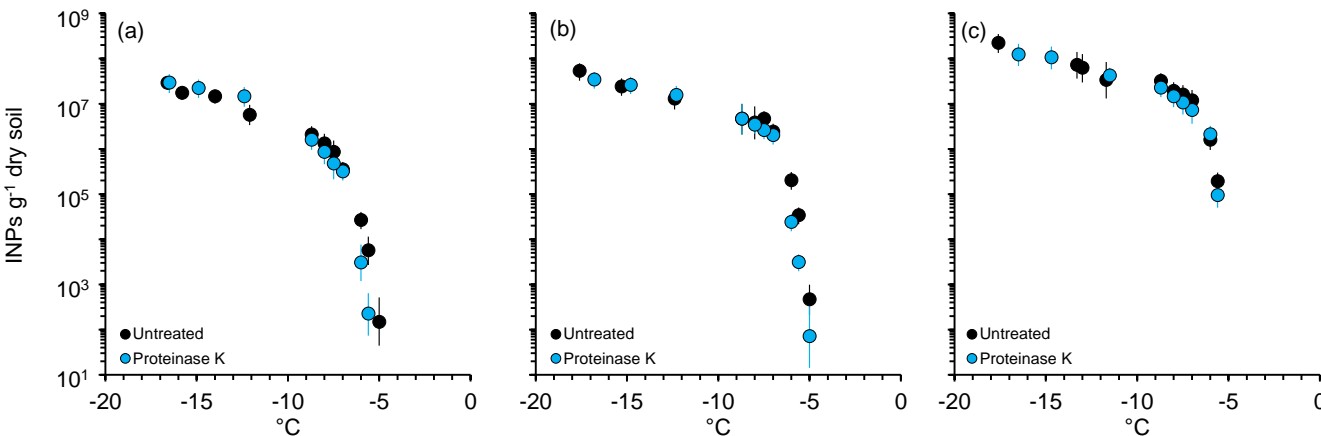

Fig. 9. Effect of Proteinase K digestion upon INPs in soils. Soils are (a) sugar beet, (b) organic grass/alfalfa fallow and (c)
pasture.





Fig. 10. Ice nucleating particle, INP1, isolated from sagebrush soil (a). After six freeze-thaw cycles it fragmented into many small pieces, the larger fragments of which are shown (b) creating at two INPs (ie, two droplets froze). Ice nucleating particle, INP2 in white light (c) and illuminated using differential interference contrast microscopy (d).





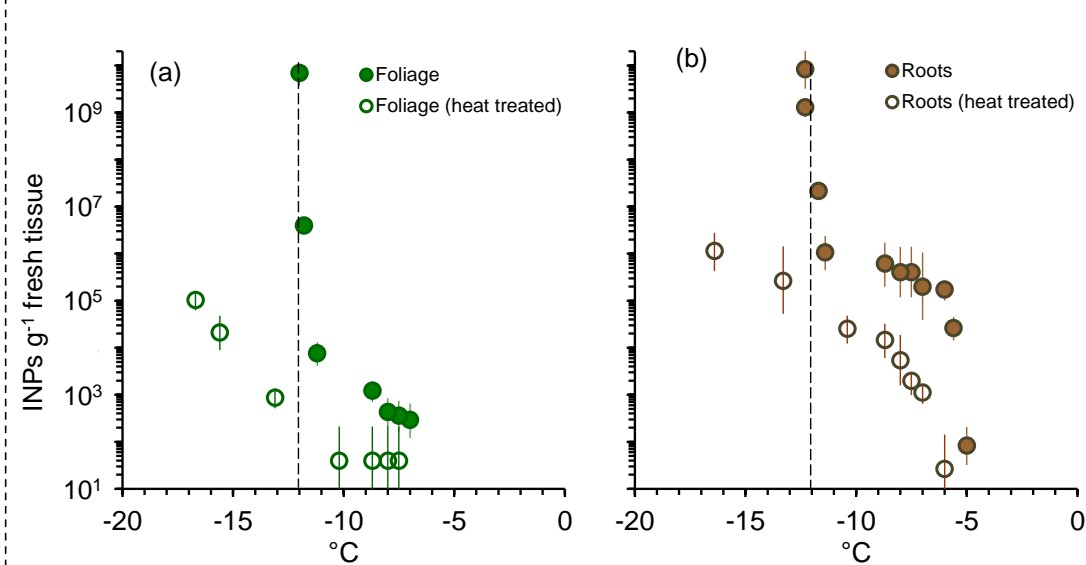

Fig. 11. Ice nucleating particles in sagebrush shoot (a) and root (b) tissues ground under liquid nitrogen and after heating to 105 °C. The dashed line at -12 °C is a guide to indicate the temperature of freezing of sagebrush tissues.



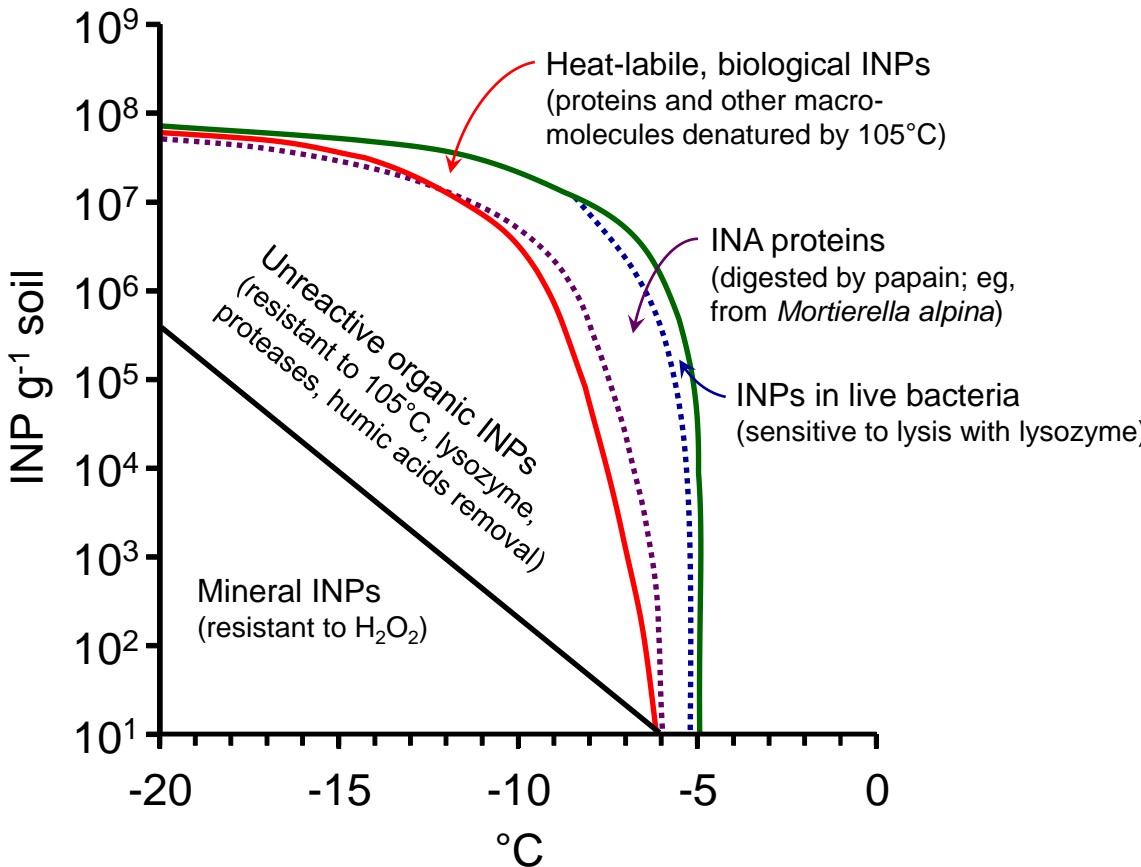

Fig. 12. Sources of ice nucleating particles in the pasture soil.