# Peer review of "Sources of organic ice nucleating particles in soils"

_Atmospheric Chemistry and Physics, 2016_

## Editor Comment (EC1) · JA Huffman (Editor) · 21 Jan 2016

Authors,

I was contacted by one of the referees who wishes to anonymously ask a technical question before proceeding with the review:

It is not clear what sizes of soil particles analyzed for INPs. Page 3, Line 19 mentions sieving soil at the 0.5 mm level, but as written, this statement refers only to the CHN analysis. Can you clarify the size range of material used for INP analysis and how this was achieved?

Thank you.

Alex Huffman
* * *

---

## Short Comment (SC1) · 25 Jan 2016

Dear Alex,

Since the aim of the work was to probe the sources of all INPs in soils (which may subsequently become associated with a range of soil particle types and sizes) we didn't sieve the soils before testing. A representative amount of each sample was initially made into a loose slurry (a 1 in 10 mix with the buffer) and shaken vigorously for 20 min to disperse it. Only samples prepared for CHN analyses were sieved, in order to obtain a fine-grained sample so that a few mg could be transferred to the aluminum vial.

Cheers, Tom

---

## Referee Comment (RC1) · P. Amato (Referee) · 6 Feb 2016

Biological ice nucleating particles (INP) in the atmosphere have raised the attention in the last decades for their probable implication in cloud freezing and precipitation. Among the different sources of primary aerosols, soils could contribute to the release of biological INPs.

The aim of this study is clearly presented: decipher the nature of INPs in soils. To address this, several soil samples were subjected to a range of treatments targeting different types of biological material: heating at 60°C and 105°C (heat-labile proteins), H2O2 (total organic matter), acid (humics), chloroforme (sterols and aliphatics), enzymes (lysozyme against bacterial IN activity, papain protease and proteinase K against proteins), the rest, resistant to all treatments, was considered mineral. The abundance of INP in treated and in untreated samples was measured by immersion freezing assays. The results show that ice nucleation is largely dominated by bio-

logical material at temperatures > -9°C, with different types of structures contributing: bacteria, proteins and other macromolecules, and others unidentified compounds recalcitrant to most treatments.

In parallel, the authors looked for the presence of known IN bacteria in the soils by targeted PCR, but failed to detect any copy of the Ina gene partly due to PCR inhibition by the soil matrices. In addition, serial dilutions of samples that froze during IN assays were done for attempting to isolate and observe by microscopy the particle from which ice was likely originally formed.

This is an interesting study bringing new information about INPs in soils. It gathers many extraction techniques for selectively remove various types of organic compounds and try to determine the relative contribution of each to the global ice nucleation capacity of soil samples. It is well written, well constructed, potential weaknesses are discussed in each subsection, and it is well referenced.

I have a few more or less major/minor comments and criticisms that I hope the authors can answer for finalizing the manuscript.

First, I found the term "source" quite misleading at some places in the manuscript (including the title) as it generally refers to a process, a geographical area or something related to it (category of landscape or else). Here it refers to different families of molecules of unidentified origin, for most, so the word "nature" rather than "source" would be more appropriate, to my opinion. Then, one of my concerns is about sample storage. It is mentioned that these were kept at 4°C, but for how long? Could the biological content of the samples have been modified during storage? The range of treatments used for targeting different classes of organic material is interesting, but probably a bit too affirmative concerning the actual efficiency and specificities. My main point here is the different treatments are presented as quite specific, i.e. targeting very narrow families of molecules, but they probably also alter untargeted (organic or mineral) molecules or incompletely remove those targeted, and this is not always really
discussed.

Concerning H2O2 treatment, it is said in the abstract "Removal of SOM with H2O2 effectively removed all INPs active >-18 °C", which is obviously not right, or I am missing something, looking at Figure 2 (some of the treated samples were frozen by this temperature). Furthermore, the method involves 15% H2O2 and boiling for 1 day. Since there is no mention of it, I assume this was done in the dark in the absence of UV light. How was it determined that "all SOM and then all residual H2O2 was decomposed"? (page 4, line 9).

Hydrogen peroxide treatment is used as a procedure for degrading OM in soils. However, a treatment at 15% H2O2 followed by 30% H2O2 at 70°C for 1-3 days only removes ∼80-90% of it (Leifeld and Kögel-Knabner, 2001). Even worst: there is sometimes still more than 50% of the original OM left in soils after 20-40 days treatment at 30% H2O2 (Eusterhues et al., 2005). The action of H2O2 on OM oxidation is based on the production of OH radicals, which requires the presence of catalysts like UV, O3 or iron for example (Kitis and Kaplan, 2007; Matilainen and Sillanpää, 2010). So, without addition of such catalysts in your samples along with H2O2, the efficiency of removal is totally dependent on the intrinsic chemical properties of your soils. Hence, I am wondering how much OM is left in your treated samples. In any case, you cannot affirm that OM is removed completely from the soils, and should acknowledge on the fact that a fraction of your "mineral INP" in figure 12 is actually probably still organic. I think that this is consistent with the fact that "mineral INPs", as named in fig 12, start inducing freezing at as warm as -6°C while this is generally not observed with minerals tested pure. If you still have some samples of your soils, it would be interesting to determine how much OM is left after such H2O2 treatment, and what is the iron content in these samples.

Heat-treatment is currently widely used as a method for suppressing proteinaceous IN activity. It is mentioned in the conclusion that it may also modify the IN ability of crystals of organic material by dissolving them. Concerning mineral crystals, the reference cited
(Zolles et al. 2015) indeed reports a little effect of heating at 250°C on the IN activity of feldspar due to surface modification. To me this is a different phenomenon which does not attest of the absence of mineral dissolution in your samples. Also, I do not get why the lack of effect of heating at 60°C demonstrates that this was not the case at 105°C (page 14, line 14). Can you clarify this?

Acid treatment for removing humic and fulvic acids: a method employed by the International Humic Substance Society was used. It involves concentrated HCl and further neutralization with NaOH. Even if the soils were decanted upon treatment, concentrated HCl and NaOH have probably modified the ionic strength of the samples (these were apparently not rinsed), and I am wondering to which extent this affected the results observed. Could you give information on that? Acid treatment also denaturate many other organic molecules than humic and fulvic acids (as mentioned page 9 line 24), and it also probably solubilizes metals (Snape et al., 2004) which are known IN (Chen et al., 1998; Phillips et al., 2008). Maybe metals are comprised into the definition of "fulvics" given page 9 line 6? This should at least be mentioned and discussed. This is also relevant since, independently of their IN activity, metals like iron are suspected to be complexed with HULIS in the atmosphere (e.g., Parazols et al., 2006).

Similarly, chloroform extracts lipids and so it is likely to inactivate IN due to bacteria. Results shows that chloroform treatment had no significant effect so suggesting to me that bacterial IN was not significant in the samples, also confirmed by PCR approach. However, lysozyme had little effect, leading to the conclusion that bacterial INPs were present but below the detection limit of PCR. Do you have evidence that bacterial INP could resist chloroform treatment? If not, how can you explain that chloroform did not affect bacteria (discrepancy between chloroform and lysozyme treatments)? About the PCR products showed in figure 7, it seems to me that a band corresponding to the gene targeted was actually present in the pasture soil (the gel is not completely horizontal in the image). How can you affirm this was not the right band? Have you any other information not mentioned in the text that helped you decide?

The exploratory method attempted for isolating particles by dichotomy and observe them, is new, to my knowledge, and I found the idea quite interesting for further investigations. Just for this reason it deserves to be presented here. However, at this stage of development this did not bring much information about soil INPs, except these are indeed particles and that they are aggregates of multiple unidentified compounds. It was to me mainly "recreational" in the paper, and probably a bit too affirmative, with no evidence for it, that the particle observed was indeed the INP (notably the first sentence page 12 line 28).

Fig 12 is an interesting summary of the results for the pasture soil, but it needs to be completed and probably modified. First, this sample was not subjected to chloroform, so the conclusion that the refractory biological IN are not removed by chloroform in this sample (page 15, line 4) is obviously erroneous. Also, considering my comments about H2O2 treatment, the mineral fraction should be even smaller, or uncertainties somehow indicated. Finally, legend is incomplete: why are there dotted and straight lines? What do represent the green line? And it is not clear to me (although I can guess) where is the "notable segment of organic INPs active below -10 to -12 °C that was unaffected by any challenge short of oxidation with H2O2" mentioned in the text. Perhaps I would be a good idea to indicate it on the Figure.

Typing and references errors: - Page 6, line 2: some words are missing in this sentence. - Bigg et al. 2015, Wright et al. 2015 (page 2 line 16), O'Sullivan et al. 2011 (page 2 line 22) and Popovitz et al. 1994 (page 10 line 2) - are missing in the list of references; - Check Tobo et al. (2104!) (page 4 line 15) and Zolles et al (2105!) (Page 6, line 17); - Pouleur et al. 92 should be 1992 (page 5 line 19); - Gavish et al. 1980 should be 1990 (page 18); - Balch et al. 2013 (page 16), DeMott and Prenni 2010 (page 17), Hayes et al. 2001 (page 18), Rigg et al. 2013 (page 22) and Wagenbrenner et al. 2013 (page 23) are not cited in the text.

References cited:

Chen, Y., Kreidenweis, S. M., McInnes, L. M., Rogers, D. C. and DeMott, P. J.: Single particle analyses of ice nucleating aerosols in the upper troposphere and lower stratosphere, Geophys. Res. Lett., 25(9), 1391–1394, doi:10.1029/97GL03261, 1998.

Eusterhues, K., Rumpel, C. and Kögel-Knabner, I.: Stabilization of soil organic matter isolated via oxidative degradation, Organic Geochemistry, 36(11), 1567–1575, doi:10.1016/j.orggeochem.2005.06.010, 2005.

Kitis, M. and Kaplan, S. S.: Advanced oxidation of natural organic matter using hydrogen peroxide and iron-coated pumice particles, Chemosphere, 68(10), 1846–1853, doi:10.1016/j.chemosphere.2007.03.027, 2007.

Leifeld, J. and Kögel-Knabner, I.: Organic carbon and nitrogen in fine soil fractions after treatment with hydrogen peroxide, Soil Biology and Biochemistry, 33(15), 2155–2158, doi:10.1016/S0038-0717(01)00127-4, 2001.

Matilainen, A. and Sillanpää, M.: Removal of natural organic matter from drinking water by advanced oxidation processes, Chemosphere, 80(4), 351–365, doi:10.1016/j.chemosphere.2010.04.067, 2010.

Parazols, M., Marinoni, A., Amato, P., Abida, O., Laj, P. and Mailhot, G.: Speciation and role of iron in cloud droplets at the puy de Dôme station, J Atmos Chem, 54(3), 267–281, doi:10.1007/s10874-006-9026-x, 2006.

Phillips, V. T., DeMott, P. J. and Andronache, C.: An empirical parameterization of heterogeneous ice nucleation for multiple chemical species of aerosol, Journal of the Atmospheric Sciences, 65(9), 2757–2783, 2008.

Snape, I., Scouller, R. C., Stark, S. C., Stark, J., Riddle, M. J. and Gore, D. B.: Characterisation of the dilute HCl extraction method for the identification of metal contamination in Antarctic marine sediments, Chemosphere, 57(6), 491–504, doi:10.1016/j.chemosphere.2004.05.042, 2004.

---

## Referee Comment (RC2) · R. Schnell (Referee) · 6 Feb 2016

In addition to the excellent review by P.Amoto I have the following comments.

General Comments: This paper presents the results of a wide range of chemical and heat stress tests on a small selection of soils from Wyoming and Colorado to determine the number and nature of the organic ice nucleating (IN) particles in the soils. The paper has above average detail in referencing earlier research in the field, the methods are sufficiently documented and the paper is well written.

Overall the paper presents a unique set of data that expands knowledge on organic IN in soils. It is an important paper and will be a well referenced publication.

Specific Comments:

Page 3: lines 17 and 18. Since the 0.1 M HCL contained a lot of water, large numbers of organic IN may have been in the wash that was discarded. Were the washes tested

for IN content? How is this potential large removal of IN taken into account? What is the purpose in sieving the soil? Were the smaller or larger particles used in the follow-on analyses?

Page 4: lines 10 and 11: Why were the samples filtered?

Page 4: lines 20-23: Again the water that was discarded probably had more organic IN than those left in the soil residue. Were the filtrates tested? Do I not understand the process here?

Page 7: lines 29-31: Same issue as above. Most organic IN will be in the filtrate, not within the remaining soil.

Page 8: lines 10-13: Schnell and Vali focused on litters from deciduous trees as they were noted to have more and better IN than evergreen vegetation. Possibly the earlier sample of evergreen litter contained deciduous plant litter as in later collections done more carefully, the evergreen litters were found to be poorer sources of IN than deciduous plant litters.

Page 10: line 6: When this test was repeated some years later it was found that the original tests probably allowed fine particles of dried leaf litter to contaminate the collecting apparatus. As such, vapor transfer of IN to the atmosphere should be discounted.

Page 11: line 12-13: It is not required that bacteria be alive to have IN activity as referenced in a number of publications cited in this paper. The IN on dead bacteria are general not the most active (no activity in the -1.5 to -3O C range) but can be active at colder temperatures.

Technical Corrections:

I read the paper mainly for the science. Dr. Amato has done an excellent job in the technical details.

---

## Referee Comment (RC3) · Anonymous Referee #3 · 8 Feb 2016

This paper is an interesting and useful contribution, generally well done and clearly written. There is little I can remark in addition to what has already been said in the two earlier reviews.

Abstract, last sentence: What do you mean by "…protected by different mechanisms."? Perhaps it should say something like "… and susceptible to different forms of stress."? None of the treatments applied in the study did test protective mechanisms, they all targeted weak points in certain groups of INPs, from which protective mechanisms can not necessarily be deduced.

Page 5, line 2: At least some of the INP fraction < 0.45 um must have been lost from the sample during chloroform treatment. Did you process controls in the same manner without chloroform?

Page 11, line 28: Here, the reader could be referred to another study currently dis-

cussed in ACPD (http://www.atmos-chem-phys-discuss.net/acp-2015-1018/), exactly showing the mentioned possibility.

Page 12, lines 13-14: Does the statement about fungal species having very sporadic occurrences of vegetative growth before dying away also apply to saprophytic soil fungi, such as M. alpina? I would have thought they have a more steady resource supply, hence vegetative growth. Crop residues of the preceding season are found in most cropped soils at least until new residues are supplied by a new crop.

Fig. 2: The almost perfect complementation on the colder side of the bulk soil spectra in Fig. 2a and 2b by data from Tobo et al. (2014) requires additional discussion. Data by Tobo et al. (2014) relates to particles of 0.6 um diameter, probably with a much larger specific surface area than that of the bulk soils studied here. Expressed in terms of INPs per gram dry soil, I would therefore expect much smaller numbers for the bulk soil, compared to those of particles of 0.6 um diameter. The good correspondence between the two sets of data might be due to several biases fortuitously cancelling each others effects.

---

## Referee Comment (RC4) · D. O'Sullivan (Referee) · 8 Feb 2016

In this paper, Hill et al. detail their efforts to unravel the sources of biogenic INPs in soils collected from Wyoming and Colorado. The authors employ a variety of extraction techniques, chemical/biochemical and physical tests to elucidate the origins of the biological INPs in the soil samples. The paper is an important piece of work, generally well-written and yields new insights into the relative contributions of different sources to the reservoir of INPs in soils. I recommend that it be published following consideration of the comments below:

Main Comments:

Page 3 lines 8-20: Am I correct in assuming that the soils weren't sieved at all before INP assays were performed? While roots were removed, what about gravel and stones? How could this subsequently impact upon calculation of INPs per gram soil?
[Figure]

Page 3 line 14/15: There is some ambiguity here. At line 14 it says that soils were stored at 4 °C before testing. Then, in the next sentence, it is mentioned that sub-samples were frozen at -20°C for "later analyses". Was INP analysis one of the later analyses specified here? If so, the effects of freezing on the soils (e.g. promotion of dessication, lysing microbial cells and effects of SOM structure) are worth noting. Additionally, were the samples air dried before freezing?

Page 3, line 23: On the "appropriate INP-free diluent", what was this? I may have just missed this, but I can't seem to find what this was for untreated soils (e.g. figure 1). Were there any runs performed where just water was used? Were there any differences observed between when a buffer was used, versus when it wasn't for untreated soils?

Page 3, line 25-26: Could the authors state what the temperature ramping profile used here was (rather than the reader having to consult Garcia et al.). It might also be appropriate to discuss how ramp rate could potentially lead to differences if others were to conduct similar style experiments but with different ramping profiles.

Page 4, line 9: What was the pH restored with?

Page 5, line11-21: Adsorption of proteins to mineral surfaces can affect their suscepti-bility to thermal denaturation (some examples: Steadman et al., 1992;Rao et al., 2000)- probably worth a mention of this here.

Page 8, lines 18-23: Line 19 states that O'Sullivan (2014) used a gentler peroxide treatment than used here. Conversely, this paper is subsequently referenced at line 23 noting that peroxide has no effect on K-feldspar. But the treatment was different from that used in this paper? With the peroxide treatment used in this paper, does it have an effect on K-feldspar? Also, for the effects of peroxide on montmorillonite, how do the peroxide treatments used by Conen et al. compare to those used here?

Page 8, line 25-28: The similarity in the results from aerosolized dusts from Tobo (2014) to the drop freezing assays here is interesting, but despite the very different techniques

used (0.6 $\mu$m vs. unsieved soils) no discussion of potential caveats is presented here. Between both techniques, very different populations of particles are being probed, yet there's no discussion of this. There's no reason to assume that active site densities for 0.6 $\mu$m and unsieved soils will continue to overlap over the entire temperature range of the plot. For the avoidance of ambiguity, I suggest discussing this point.

Typos: Page 6, line 2: Seems to be a typo here- "from (grown in nutrient broth. . .")

References: Rao, M. A., Violante, A., and Gianfreda, L.: Interaction of acid phosphatase with clays, organic molecules and organo-mineral complexes: kinetics and stability, Soil Biol. Biochem., 32, 1007-1014, 2000.

Steadman, B. L., Thompson, K. C., Middaugh, C. R., Matsuno, K., Vrona, S., Lawson, E. Q., and Lewis, R. V.: The effects of surface adsorption on the thermal stability of proteins, Biotechnol. Bioeng., 40, 8-15, 1992.

---

## Author Comment (AC1) · 1 May 2016

**Sources of organic ice nucleating particles in soils**

T. C. J. Hill, P. J. DeMott, Y. Tobo, J. Fröhlich-Nowoisky, B. F. Moffett, G. D. Franc and S. M. Kreidenweis

**D. O'Sullivan (Referee)**
We thank Dr O'Sullivan for his supportive overview remarks.

*Page 3 lines 8-20: Am I correct in assuming that the soils weren't sieved at all before INP assays were performed? While roots were removed, what about gravel and stones? How could this subsequently impact upon calculation of INPs per gram soil?*
By chance, no soils used in this work had any gravel-sized stones (since they were mostly forest humic horizons and fine alluvial silts). In two, the native grassland and sagebrush, there were some medium-sized stones and cobbles, which were excluded during sampling. A clarification will be added to the Methods.

*Page 3 line 14/15: There is some ambiguity here. At line 14 it says that soils were stored at 4 C before testing. Then, in the next sentence, it is mentioned that subsamples were frozen at -20 C for "later analyses". Was INP analysis one of the later analyses specified here? If so, the effects of freezing on the soils (e.g. promotion of desiccation, lysing microbial cells and effects of SOM structure) are worth noting. Additionally, were the samples air dried before freezing?*
The frozen subsamples were stored for potential future tests, and in fact were used for papain digests (in addition to DNA), since the susceptibility of the *Mortierella alpina* INPs to papain was discovered later by our collaborators at Max Planck, Mainz. This will be stated in the methods, as will the fact that they were stored in 15 ml screw-cap tubes (ie, desiccation resistant).
Re the effects of freezing, these soils all naturally experience repeated freeze-thaw cycles in colder seasons, and so we doubt that there would have been any effect on SOM that they hadn't already experienced. Fortunately too, the INPs we were testing for in these samples were expected to be cell-free proteins, and so lysis of the microbiota shouldn't have had a major impact. However, we will add a note in the Results section concerning papain that these were test on frozen samples, and so may not be completely comparable with fresh soil.

*Page 3, line 23: On the "appropriate INP-free diluent", what was this? I may have just missed this, but I can't seem to find what this was for untreated soils (e.g. figure 1). Were there any runs performed where just water was used? Were there any differences observed between when a buffer was used, versus when it wasn't for untreated soils?*
The INP-free diluent was phosphate buffer for standard tests, but Tris/EDTA was required for lysozyme and proteinase K. Presence of EDTA is especially important for lysozyme activity, as noted in the text, since it chelates divalent cations which, by a mechanism that is still not fully understood, creates small holes in the outer membranes of the Gram negative bacteria enabling the enzyme to penetrate and digest the underlying wall. Tris was also used for both of these enzymes since it is the default buffer used in their testing, and so was judged likely to

maximize their activity. To avoid confusion we have removed the word "appropriate" from this section of the text, since it suggests that yet more buffers were used.

No, we didn't compare the use of buffer (typically pH adjusted to approximately match each soil's state) versus deionized water for untreated samples. The rationale for this was two-fold. Firstly, if we had used deionized water, which is always slightly acid, we would have changed the pH of the soil suspensions of the more alkaline soils with progressive dilutions, and so perhaps have introduced a pH-change-driven bias. Secondly, deionized water is not a natural state for a soil solution, and indeed strongly strips ions off mineral particles, which we now know can deactivate K-Feldspar, for example. Hence, we opted to use dilute buffers. We did compare the effect of using PO4 buffer with Tris/EDTA and found no systematic differences (see attached Figures in the file "INP spectra in two buffers).

*Page 3, line 25-26: Could the authors state what the temperature ramping profile used here was (rather than the reader having to consult Garcia et al.). It might also be appropriate to discuss how ramp rate could potentially lead to differences if others were to conduct similar style experiments but with different ramping profiles.*

We have added a summary of the method, and mentioned how this might change slightly the results (ie, give higher results due to the longer dwell times) in comparison to other methods.

*Page 4, line 9: What was the pH restored with?*

My notes show that only one soil, the SAREC pasture, had a significantly different pH after peroxide digestion. It fell from 7.8 to 6.8, and was restored with drop-wise addition of 1 M NaOH. The kaolinite increased in pH from 4.2 to 5.6 and was restored similarly with 1 M HCl.

*Page 5, line11-21: Adsorption of proteins to mineral surfaces can affect their susceptibility to thermal denaturation (some examples: Steadman et al., 1992; Rao et al., 2000) - probably worth a mention of this here.*

These papers show that while adsorption generally decreases the thermal stability of proteins, the effect can be both to increase stability or increase their sensitivity to denaturation to 60°C heat. We will mention this possibility and cite Rao et al. (2000) since they tested an acid-phosphatase, common and important in soil processes, with both clay and tannins, whereas Steadman et al. (1992) tested mainly proteins from animals.

*Page 8, lines 18-23: Line 19 states that O'Sullivan (2014) used a gentler peroxide treatment than used here. Conversely, this paper is subsequently referenced at line 23 noting that peroxide has no effect on K-feldspar. But the treatment was different from that used in this paper? With the peroxide treatment used in this paper, does it have an effect on K-feldspar? Also, for the effects of peroxide on montmorillonite, how do the peroxide treatments used by Conen et al. compare to those used here?*

O'Sullivan et al. (2014) and Conen et al. (2011) both used 35% $H_2O_2$, whereas we used only a 15% solution. By contrast, O'Sullivan heated their samples to 50°C, whereas we boiled ours. 35% peroxide is a very powerful oxidizer, and so we think it is fair to judge that if exposure of K-feldspar to 35% at 50°C had no effect on its activity then it probably would not be altered by 15% at roughly 105°C (the boiling point when adjusted for the altitude). (O'Sullivan et al. 2015) show no effect of heating K-feldspar to 95°C in water.) Conen do not provide any details about their digestion protocol other than that they used 35% peroxide.

*Page 8, line 25-28: The similarity in the results from aerosolized dusts from Tobo (2014) to the drop freezing assays here is interesting, but despite the very different techniques used (0.6 μm vs. unsieved soils) no discussion of potential caveats is presented here. Between both techniques, very different populations of particles are being probed, yet there's no discussion of this. There's no reason to assume that active site densities for 0.6 μm and unsieved soils will continue to overlap over the entire temperature range of the plot. For the avoidance of ambiguity, I suggest discussing this point.*

The two methods corresponded quite well, although, since they span 10 orders of magnitude, the differences are also minimized. A closer look at Fig2a and 2b suggests that the results from Tobo et al. (2014) are 3-4x higher than found by immersion freezing tests of the bulk agricultural soil, and that for kaolinite the difference is likely wider. But this is a valid point, and the similarities may be due as much to a fortuitous choice in conversion (eg, an assumption of sphericity of the 0.6 μm particles). We have, therefore, made this point, and emphasized that the aspect of particular interest is the reduction in INPs after peroxide digestion in both sets of data, not their absolute values.

*Typos: Page 6, line 2: Seems to be a typo here- "from (grown in nutrient broth. . .")*
Corrected.

**Anonymous Referee #3**
We thank Referee #3 for their supportive and generous general remarks.

*Abstract, last sentence: What do you mean by ". . .protected by different mechanisms."? Perhaps it should say something like ". . . and susceptible to different forms of stress."? None of the treatments applied in the study did test protective mechanisms, they all targeted weak points in certain groups of INPs, from which protective mechanisms can not necessarily be deduced.*

Referee #3 makes a very good point. Since the tests performed in this paper are all non-natural, it may be inappropriate at this stage to make any judgments about their susceptibility to various stressors. We have thus removed this point from the last sentence of the Abstract.

*Page 5, line 2: At least some of the INP fraction < 0.45 um must have been lost from the sample during chloroform treatment. Did you process controls in the same manner without chloroform?*

The chloroform did not really generate a soil suspension in the classic sense, as water would have. Indeed it remained very clear, apart from some slight yellowing, and after swirling would settle immediately. The filtering took just a few seconds, suggesting there was no effective suspension of fines. We did process controls, yes, using buffer. That is, the unamended spectra in the figures is the same dried soil sample suspended in buffer. We could have filtered the filtrate through an even finer pore size and tested that. However, there was clearly negligible losses; as shown in Fig. 4 there was no significant difference in INPs before and after the chloroform extraction.

*Page 11, line 28: Here, the reader could be referred to another study currently discussed in ACPD ([http://www.atmos-chem-phys-discuss.net/acp-2015-1018/](http://www.atmos-chem-phys-discuss.net/acp-2015-1018/) ), exactly showing the mentioned possibility.*

Indeed. I read the mentioned paper after I submitted this work, and agree completely. It has been added.

*Page 12, lines 13-14: Does the statement about fungal species having very sporadic occurrences of vegetative growth before dying away also apply to saprophytic soil fungi, such as M. alpina? I would have thought they have a more steady resource supply, hence vegetative growth. Crop residues of the preceding season are found in most cropped soils at least until new residues are supplied by a new crop.*

Yes, Warcup's unique study inherently isolated saprophytic fungi (he transferred individual hyphae onto agar). But, yes, that is correct about these soils; they did have degradable organic matter dispersed through the A horizon, although not uniformly of course. Even so, it is well known that dilution plating heavily favours spore formers, and, hence, the relative abundance of M. alpina found by Fröhlich-Nowoisky et al (2015) using dilution plating need not correlate at all with vegetative or cell-free INP abundance. We have, however, modified that paragraph to accommodate the Reviewer's comment.

*Fig. 2: The almost perfect complementation on the colder side of the bulk soil spectra in Fig. 2a and 2b by data from Tobo et al. (2014) requires additional discussion. Data by Tobo et al. (2014) relates to particles of 0.6 um diameter, probably with a much larger specific surface area than that of the bulk soils studied here. Expressed in terms of INPs per gram dry soil, I would therefore expect much smaller numbers for the bulk soil, compared to those of particles of 0.6 um diameter. The good correspondence between the two sets of data might be due to several biases fortuitously cancelling each others effects.*

As discussed above in the response to Dr O'Sullivan, a closer inspection of Tobo et al.'s work indicates their results do seem 3-4x higher than obtained using immersion freezing with the bulk soil. As also raised by Daniel O'Sullivan, we agree that the correspondence may be as much coincidence as real, and so will include the well-expressed phrase "due to several biases fortuitously cancelling each others effects" in the text.

**R. Schnell (Referee)**
We appreciate the overview very much, thank you.

*Page 3: lines 17 and 18. Since the 0.1 M HCL contained a lot of water, large numbers of organic IN may have been in the wash that was discarded. Were the washes tested for IN content? How is this potential large removal of IN taken into account? What is the purpose in sieving the soil? Were the smaller or larger particles used in the follow-on analyses?*

This paragraph details only the preparation of samples for dry weights, and total C and N (grinding followed by 0.5 mm sieving is the standard procedure in preparing samples for CHN analyzers), and not for any INP measures. Since this is ambiguous at present, we have added a clarification.

*Page 4: lines 10 and 11: Why were the samples filtered?*

This is another clarification issue. We used 0.45 µm filtered buffer (filtered to remove any insolubles or rogue dust in the two salts used to make the buffer) as the diluent. We didn't filter the dilution series itself. The wording has been modified to hopefully make this clear.

*Page 4: lines 20-23: Again the water that was discarded probably had more organic IN than those left in the soil residue. Were the filtrates tested? Do I not understand the process here?*

This process is the standard IHSS method to remove fulvic and humic acids. Both supernatants were clear after settling. Indeed, we tested the INP spectrum of the residual soil after the 1 M HCl step and saw no reduction. Hence, although some particulate material with IN activity may have been lost by decanting, it was judged to be very minor, and so the decanted liquid itself was not tested. To emphasize this aspect as well as the lack of impact of the step to remove fulvic acids, we have included a figure illustrating the effect of 1 M HCl treatment, and referred to it in the section discussing this.

After the second step of 0.1 M NaOH treatment we only observed a reduction in INPs active ≥-7 C. As 0.1 M NaOH is extremely denaturing to proteins, it is reasonable to assume that this impact was due to destruction of IN proteins from fungi and bacteria and not losses in the supaernatant, which would have had a broader impact across a range of INPs.

*Page 7: lines 29-31: Same issue as above. Most organic IN will be in the filtrate, not within the remaining soil.*

We don't' agree. This step was an essential vigorous rinsing of intact roots and shoots to remove attached soil, with the specific aim of removing as much of this attached material in order to specifically test the plant material. But we have added a clarification to make clear that the plant tissues were the focus of testing.

*Page 8: lines 10-13: Schnell and Vali focused on litters from deciduous trees as they were noted to have more and better IN than evergreen vegetation. Possibly the earlier sample of evergreen litter contained deciduous plant litter as in later collections done more carefully, the evergreen litters were found to be poorer sources of IN than deciduous plant litters.*

We have included this note.

*Page 10: line 6: When this test was repeated some years later it was found that the original tests probably allowed fine particles of dried leaf litter to contaminate the collecting apparatus. As such, vapor transfer of IN to the atmosphere should be discounted.*

Thank you; we have removed this observation.

*Page 11: line 12-13: It is not required that bacteria be alive to have IN activity as referenced in a number of publications cited in this paper. The IN on dead bacteria are general not the most active (no activity in the -1.5 to -3O C range) but can be active at colder temperatures.*

This is very true. Hence we have added a note to this effect.

**P. Amato (Referee)**

Thank you, to Dr Amato, for his encouraging overview. Also, as noted by Russell Schnell, Dr Amato has raised some carefully-considered and valid questions about the work and its interpretations, which we hope are properly addressed and adjusted for, as detailed below.

*First, I found the term "source" quite misleading at some places in the manuscript (including the title) as it generally refers to a process, a geographical area or something related to it (category of landscape or else). Here it refers to different families of molecules of unidentified origin, for most, so the word "nature" rather than "source" would be more appropriate, to my opinion.*

The Oxford Dictionary defines one of the meanings of source to be "origin, chief or prime cause". This work wasn't able to pinpoint the exact nature/class of molecules acting as INPs in many cases, such as the organic INPs active below -10° to -12 °C, but we were able to identify several pools as sources of INPs. So, we think that the word is suitable for the title.

Having said that, if Dr Amato finds it misleading then other readers will too. We have, therefore, checked every occurrence of "source" in the text, and in order to refine its meaning to be only that of a circumscribed group of organic molecules from which a class of INPs originates, and have replaced other uses of the word with "identity", "nature", "class", "composition", and "group".

*Then, one of my concerns is about sample storage. It is mentioned that these were kept at 4 C, but for how long? Could the biological content of the samples have been modified during storage?*

Samples were stored at 4°C in bags that were vented to allow gas exchange and analyzed within 1-2 weeks. Indeed, in order to make sure we only used fresh material we re-sampled from the field to obtain a second batch of fresh soil, for most soils (see Table 1). The baseline INP spectra of the soils also appeared to be very stable (see attached Figures in file "INP spectra in two buffers), a characteristic that was the subject of an Abstract submitted to AGU in 2015 (see attached).

*The range of treatments used for targeting different classes of organic material is interesting, but probably a bit too affirmative concerning the actual efficiency and specificities. My main point here is the different treatments are presented as quite specific, i.e. targeting very narrow families of molecules, but they probably also alter untargeted (organic or mineral) molecules or incompletely remove those targeted, and this is not always really discussed.*

This is a very good point. In the concluding section we had discussed this aspect in relation to heat treatments and minerals: "Immersion in hot water could also affect the IN ability of minerals. For example, the IN activity of K-feldspar (Atkinson et al., 2013) may be altered by a change in composition of surface ions (Zolles et al., 2015)."

However, we had overlooked doing the same for enzymes. Hence, we have added the following in that same section: "The three enzymes have very specific sites of activity within their target molecules (lysozyme also hydrolyses fungal chitin oligosaccharides, but doesn't digest the chitin polymer itself), but may have also have had non-target influences. For example, enzymes may bind to and block INPs on mineral surfaces (see Zolles et al., 2015). Further, while reaction conditions for all enzymes were theoretically optimal, and ample time given for completion, their efficiencies aren't guaranteed in such a complex milieu as soil; their substrates

may have been protected by adsorption to clays (O'Sullivan et al., 2016) or tannins or by location within flocs or biofilms, and they themselves may have been inactivated by similar mechanisms or by cleavage by native proteases."

As detailed below, we have also added cautions about the effectiveness and impact of the $H_2O_2$ digestions.

*Concerning $H_2O_2$ treatment, it is said in the abstract "Removal of SOM with $H_2O_2$ effectively removed all INPs active >-18 C", which is obviously not right, or I am missing something, looking at Figure 2 (some of the treated samples were frozen by this temperature).*

We have changed the sentence to "removal of SOM with $H_2O_2$ removed ≥99% of all INPs active >-18°C".

*Furthermore, the method involves 15% $H_2O_2$ and boiling for 1 day. Since there is no mention of it, I assume this was done in the dark in the absence of UV light. How was it determined that "all SOM and then all residual $H_2O_2$ was decomposed"? (page 4, line 9). Hydrogen peroxide treatment is used as a procedure for degrading OM in soils. However, a treatment at 15% $H_2O_2$ followed by 30% $H_2O_2$ at 70 C for 1-3 days only removes 80-90% of it (Leifeld and Kögel-Knabner, 2001). Even worst: there is sometimes still more than 50% of the original OM left in soils after 20-40 days treatment at 30% $H_2O_2$ (Eusterhues et al., 2005). The action of $H_2O_2$ on OM oxidation is based on the production of OH radicals, which requires the presence of catalysts like UV, O3 or iron for example (Kitis and Kaplan, 2007; Matilainen and Sillanpää, 2010). So, without addition of such catalysts in your samples along with $H_2O_2$, the efficiency of removal is totally dependent on the intrinsic chemical properties of your soils. Hence, I am wondering how much OM is left in your treated samples. In any case, you cannot affirm that OM is removed completely from the soils, and should acknowledge on the fact that a fraction of your "mineral INP" in figure 12 is actually probably still organic. I think that this is consistent with the fact that "mineral INPs", as named in fig 12, start inducing freezing at as warm as -6 C while this is generally not observed with minerals tested pure. If you still have some samples of your soils, it would be interesting to determine how much OM is left after such $H_2O_2$ treatment, and what is the iron content in these samples.*

Yes, the digestions were performed in the laboratory (ie, no UV light supplied). However, the 15% $H_2O_2$ reacted rapidly with the soils when initially added (it would froth up and overflow the beaker if not stirred), which indicated the presence of ample catalysts, presumably $Fe^{2+}/Fe^{3+}$, Mn, etc. When this initial stage had slowed, the temperature was raised to boiling and the process continued vigorously (small flecks of plant material, for example, could be seen fizzing). This was maintained for 2-3 h until all organic material had visibly disappeared, and then the boiling was continued until it changed from an effervescent to a rolling boil, indicative of the decomposition of all residual $H_2O_2$. The beaker was then left overnight.

In one soil, the sugar beet soil, we observed the following day that the supernatant "remained cloudy", in contrast to other soils that cleared rapidly. The volume was noted, the sample stirred and an aliquot taken for testing. $H_2O_2$ was then added and the sample boiled again as normal. We noted that, this time, the suspension "quickly settled". Another aliquot was taken, and the process was repeated a third time. The results of this test are shown below. They show that indeed the residual cloudiness did indicate the presence of remaining organic material containing INPs, and that after the second digest, judged to be complete, the spectrum had

stabilized; by this we mean that no significant change occurred when the sample was treated a third time (used in the paper).

Furthermore, minerals have a log-linear spectrum of activity, in contrast to organic INP which produce a "hump" at the warm end of the spectrum, starting at -15°C. After peroxide treatment, all the residual sediments exhibited this log-linear mineral trend.

[Figure]

Figure. Effect of repeated $H_2O_2$ digestions upon IN activity of the sugar beet soil.

It is true that all oxidizers incompletely remove the SOM, so we have add the following to Methods section "(but see Mikutta et al. (2005) for a review of the inefficiencies inherent in the use of oxidizers to remove SOM from soils)".

However, although likely that the peroxide treatment was not completely effective at removing all organic matter, it did lower INP concentrations by ≥99% at all tested temperatures, and typically by >99.9%. Hence, the treatment did demonstrate that the INPs were effectively organic.

With regard to Fig. 12, while most minerals do indeed not start freezing at -6°C, K-feldspar certainly does (up to -2°C). As suggested, we have added a sentence to the legend of Fig. 12 noting and explaining why the line defining the INP spectrum of the mineral component should be considered approximate.

*Heat-treatment is currently widely used as a method for suppressing proteinaceous IN activity. It is mentioned in the conclusion that it may also modify the IN ability of crystals of organic material by dissolving them. Concerning mineral crystals, the reference cited (Zolles et al. 2015) indeed reports a little effect of heating at 250 C on the IN activity of feldspar due to surface modification. To me this is a different phenomenon which does not attest of the absence of mineral dissolution in your samples. Also, I do not get why the lack of effect of heating at 60 C demonstrates that this was not the case at 105 C (page 14, line 14). Can you clarify this?*

Yes, Zolles et al. (2015) do show that. However, 250°C is an extreme treatment. By contrast, O'Sullivan et al. (2015) show that heating K-feldspar in water to 95°C had no effect upon INP activity. To add support for our judgment that activity of K-feldspar would not be affected by immersion in hot water, either by the removal of ions or some other mechanism, we have added the latter reference to that discussion.

*Acid treatment for removing humic and fulvic acids: a method employed by the International Humic Substance Society was used. It involves concentrated HCl and further neutralization with NaOH. Even if the soils were decanted upon treatment, concentrated HCl and NaOH have probably modified the ionic strength of the samples (these were apparently not rinsed), and I am wondering to which extent this affected the results observed. Could you give information on that? Acid treatment also denaturate many other organic molecules than humic and fulvic acids (as mentioned page 9 line 24), and it also probably solubilizes metals (Snape et al., 2004) which are known IN (Chen et al., 1998; Phillips et al., 2008). Maybe metals are comprised into the definition of "fulvics" given page 9 line 6? This should at least be mentioned and discussed. This is also relevant since, independently of their IN activity, metals like iron are suspected to be complexed with HULIS in the atmosphere (e.g., Parazols et al., 2006).*

The pasture soil settled well after being left overnight, and so after decanting off the 0.1 M HCl or 0.1 M NaOH little residual liquid remained. Serial dilutions were then made, but due to turbidity the first measures were made on the 1:100 dilution. Hence the ionic strength of this suspension would have been too low to have exerted any appreciable freezing point depression.

As described in our responses to Russell Schnell, we have now added a figure showing the lack of any impact of the 0.1 M HCl treatment upon the INP spectrum of the pasture soil.

*Similarly, chloroform extracts lipids and so it is likely to inactivate IN due to bacteria. Results shows that chloroform treatment had no significant effect so suggesting to me that bacterial IN was not significant in the samples, also confirmed by PCR approach. However, lysozyme had little effect, leading to the conclusion that bacterial INPs were present but below the detection limit of PCR. Do you have evidence that bacterial INP could resist chloroform treatment? If not, how can you explain that chloroform did not affect bacteria (discrepancy between chloroform and lysozyme treatments)?*

Lysozyme had no effect on the lodgepole pine forest soil, while in the sagebrush it only reduced INPs active between -5 and -6°C. Hence there was no discrepancy for the lodgepole and only a one degree region of difference for the sagebrush. PCR did not detect INA bacteria in either sample. The chloroform-treated soils were air-dried before steeping, and so perhaps the lysozyme-sensitive entity in the sagebrush soil was protected/stabilized from the chloroform inside its partially dried bacterial "container". Good question; we're not sure.

*About the PCR products showed in figure 7, it seems to me that a band corresponding to the gene targeted was actually present in the pasture soil (the gel is not completely horizontal in the image). How can you affirm this was not the right band? Have you any other information not mentioned in the text that helped you decide?*

Dr Amato has a keen eye! We looked at that band, but upon closer inspection it has a size of ~210 bp, whereas the true band is 194 bp (see Hill et al, 2014). (The gel used (Metaphor) has excellent resolving ability, to within a few base pairs.) You can see that it lies above the 200 bp band in the ladder, whereas the correct amplicons in the +ve controls and spiked samples lie just

below it. The dashed line on the gel is also tilted to take account of the slope. Also, when the samples were spiked with a modest number of *ina* genes (Fig. 7b) this 210 bp band and most other non-specifics did not amplify; when the correct target is present it dominates the reaction and so quenches the amplification of non-specifics that amplify late in the reaction due to occasional mis-priming by the primers.

*The exploratory method attempted for isolating particles by dichotomy and observe them, is new, to my knowledge, and I found the idea quite interesting for further investigations. Just for this reason it deserves to be presented here. However, at this stage of development this did not bring much information about soil INPs, except these are indeed particles and that they are aggregates of multiple unidentified compounds. It was to me mainly "recreational" in the paper, and probably a bit too affirmative, with no evidence for it, that the particle observed was indeed the INP (notably the first sentence page 12 line 28).*

It is true that this aspect of the work was partly a demonstration of its potential. But even knowing that both INPs were organic aggregates is useful new information; they could have been fungal spores or bacteria, or mineral particles.

Gabor Vali contacted me (TH) privately about the paper, and I re-assured him that the method was robust in terms of cleanliness, rigor and number of dilutions required. (I can forward my full reply to his questions if desired). We do also state in the paper that "It is possible that the true INP was an accompanying component too small to be visible.". However, since he also thought we may be being too confident, we have changed a sentence in the Conclusions to reflect this. It is "Two probable INPs were isolated by exploiting their IN activity (initially at -7 °C). Since both were complex entities, the specific IN component was not able to be identified."

*Fig 12 is an interesting summary of the results for the pasture soil, but it needs to be completed and probably modified. First, this sample was not subjected to chloroform, so the conclusion that the refractory biological IN are not removed by chloroform in this sample (page 15, line 4) is obviously erroneous. Also, considering my comments about H2O2 treatment, the mineral fraction should be even smaller, or uncertainties somehow indicated. Finally, legend is incomplete: why are there dotted and straight lines? What do represent the green line? And it is not clear to me (although I can guess) where is the "notable segment of organic INPs active below -10 to -12 C that was unaffected by any challenge short of oxidation with H2O2" mentioned in the text. Perhaps I would be a good idea to indicate it on the Figure.*

Indeed, yes, re extending the results with chloroform from the sagebrush and lodgepole pine soils to these; that point has been deleted. We have added a comment to the figure legend indicating that $H_2O_2$ digestion is an inherently variable process. We have also indicated in the text which is the "notable segment…". Finally, we have modified the figure to make it much clearer.

*Typing and references errors:*
*- Page 6, line 2: some words are missing in this sentence.*
*- Bigg et al. 2015, Wright et al. 2015 (page 2 line 16), O'Sullivan et al. 2011 (page 2 line 22, actually **2014**) and Popovitz et al. 1994 (page 10 line 2) are missing in the list of references;*
*- Check Tobo et al. (2104!) (page 4 line 15) and Zolles et al (2105!) (Page 6, line 17);*
*- Pouleur et al. 92 should be 1992 (page 5 line 19);*
*- Gavish et al. 1980 should be 1990 (page 18);*

*- Balch et al. 2013 (page 16), DeMott and Prenni 2010 (page 17), Hayes et al. 2001 (page 18), Rigg et al. 2013 (page 22) and Wagenbrenner et al. 2013 (page 23) are not cited in the text.*

I am very thankful to Dr Amato for picking up on these sloppy errors. Our apologies. They have been corrected.

---

## Author Comment (AC2) · 1 May 2016

[revised manuscript text omitted]

**Commented [TH2]:** Gabor Vali sent me a personal review of manuscript. This modified paragraph, a qualification of the origin has been added based on his advice.

[revised manuscript text omitted]

---

## Author Comment (AC3) · 1 May 2016

See attached figure
* * *
[Figure]

[Figure]

**Fig. 1.**